# Efficient generation of a self-organizing neuromuscular junction model from human pluripotent stem cells

Alessia Urzi[1], Ines Lahmann[1,10], Lan Vi N. Nguyen[1,10], Benjamin R. Rost [2,10], Angélica García-Pérez[1], Noemie Lelievre[1], Megan E. Merritt-Garza[3], Han C. Phan[4], Gary J. Bassell [3], Wilfried Rossoll [5,6], Sebastian Diecke [7], Severine Kunz [8], Dietmar Schmitz[2,9] & Mina Gouti [1] ✉

The complex neuromuscular network that controls body movements is the target of severe diseases that result in paralysis and death. Here, we report the development of a robust and efficient self-organizing neuromuscular junction (soNMJ) model from human pluripotent stem cells that can be maintained long-term in simple adherent conditions. The timely application of specific patterning signals instructs the simultaneous development and differentiation of position-specific brachial spinal neurons, skeletal muscles, and terminal Schwann cells. High-content imaging reveals self-organized bundles of aligned muscle fibers surrounded by innervating motor neurons that form functional neuromuscular junctions. Optogenetic activation and pharmacological interventions show that the spinal neurons actively instruct the synchronous skeletal muscle contraction. The generation of a soNMJ model from spinal muscular atrophy patient-specific iPSCs reveals that the number of NMJs and muscle contraction is severely affected, resembling the patient's pathology. In the future, the soNMJ model could be used for high-throughput studies in disease modeling and drug development. Thus, this model will allow us to address unmet needs in the neuromuscular disease field.

Human pluripotent stem cells (hPSCs) have been used extensively to generate reductionist models of either spinal cord motor neurons (MNs) or skeletal muscles to study neuromuscular system disorders[1-7]. While these models are very useful for studying cell-autonomous effects, they do not reflect the complex interactions of diverse cell types such as spinal cord neurons, skeletal muscles, glia and terminal Schwann cells required to study neuromuscular junction (NMJ) pathologies. Bioengineering and co-culture approaches increased complexity and have been successfully used as alternatives to reductionist models[8-14]. However, co-culture approaches are limited by the

[1]Stem Cell Modeling of Development & Disease Group, Max Delbrück Center for Molecular Medicine in the Helmholtz Association (MDC), 13125 Berlin, Germany. [2]German Center for Neurodegenerative Diseases (DZNE), Berlin, Germany. [3]Department of Cell Biology, Laboratory for Translational Cell Biology, Emory University School of Medicine, Atlanta, GA 30322, USA. [4]Department of Pediatrics, University of Alabama, Birmingham, AL 35294, USA. [5]Department of Neuroscience, Mayo Clinic, Jacksonville, FL 32224, USA. [6]Mayo Clinic Graduate School of Biomedical Sciences, Mayo Clinic, Jacksonville, FL 32224, USA. [7]Max Delbrück Center for Molecular Medicine in the Helmholtz Association (MDC), Technology Platform Pluripotent Stem Cells, 13125 Berlin, Germany. [8]Max Delbrück Center for Molecular Medicine in the Helmholtz Association (MDC), Technology Platform Electron Microscopy, 13125 Berlin, Germany. [9]Berlin Institute of Health, NeuroCure Cluster of Excellence, Charité-Universitätsmedizin Berlin, Corporate Member of Freie Universität Berlin and Humboldt-Universität zu Berlin, Berlin, Germany. [10]These authors contributed equally: Ines Lahmann, Lan Vi N. Nguyen, Benjamin R. Rost. ✉e-mail: mina.gouti@mdc-berlin.de

necessity to derive the required cell types separately before combining them in a single culture. This imposes practical limitations but also leads to compromises in the choice of the common culture medium conditions, which in turn limits the long-term maintenance of the co-cultures. The establishment of an advanced 2D NMJ model would facilitate long-term maintenance of neuromuscular cultures, something that eluded previously established co-culture models, even when chips, scaffolds, or protein matrices were used[8,9,15,16].

The potential of neuromesodermal progenitor (NMP) cells to generate the different cell types required for functional NMJs has opened up new opportunities for investigating NMJ disorders[17]. The simultaneous generation of the spinal cord and skeletal muscle tissues from NMPs can ensure the early establishment of crucial functional interactions that stimulate their cooperative maturation. As a proof of concept, we have previously shown that hPSC-derived NMPs are sufficient for generating functional NMJs in 3D neuromuscular organoids (NMOs)[17]. Neuromuscular organoids reach 2–3 mm in size after 30 days in culture and 4–5 mm after 50 days, but because of their size, they cannot be maintained in the multiwell format (96-well or smaller) necessary for high-throughput studies.

To overcome these limitations while retaining the advantage of simultaneous development, maturation, and long-term culture of NMJs, we sought to derive an NMJ model system in 2D through an NMP intermediate state. We show that specification of NMP identity followed by timely inhibition of BMP and TGFβ signaling instructed the development of self-organizing neural and mesodermal progenitors within a short window of time (6 days). Strikingly, this organization was re-established even after passaging and cells segregated into spatially organized spinal cord neurons, skeletal muscle and terminal Schwann cells, forming functional NMJs that matured over time.

In the neuromuscular cultures, the muscle started contracting at day 50 due to the formation of functional NMJs (Supplementary Movie 2). In-depth functional characterization using optogenetic activation of spinal cord neurons and pharmacological manipulation of the NMJs showed that the synchronous contraction of skeletal muscles was instructed by the MNs. Notably, the spinal cord neurons within the NMJ model exhibit a predominant brachial identity, encompassing both median and lateral motor column MNs, interneurons, glial cells and terminal Schwann cells. As proof of principle, we used this soNMJ model to recapitulate the disease pathology observed in patients with spinal muscular atrophy (SMA). Indeed, SMA-specific neuromuscular cultures developed a severe phenotype characterized by a reduction in the number of NMJs and compromised skeletal muscle contractions resembling the clinical presentation. High-throughput analysis of the soNMJ model revealed that muscle pathology developed prior to the onset of motor neuron loss. In the future, this opens the opportunity to design novel therapeutic strategies that target early muscle pathology in SMA patients.

## Results

### Delayed dual SMAD inhibition is necessary for the concurrent development of neural and mesodermal lineages from NMPs

We used five different hPSC lines (see methods section) to explore their potential to generate a self-organizing NMJ (soNMJ) model through an NMP state. We have previously used hPSC-derived NMP cells to generate NMOs in 3D under minimal culture media. However, the direct transfer of the protocol from 3D to 2D was insufficient to generate a functional NMJ model in adherent culture conditions. Analysis of the adherent cultures after applying the NMO conditions at days 20 and 50 revealed the presence of neurons expressing TUBB3 and a few muscle progenitors expressing DESMIN at day 20, which gave rise to only a few differentiated skeletal myofibers at day 50 (Supplementary Fig. 1).

We reasoned that transposition of the 3D early NMO culture conditions in 2D may change the autocrine signal signature. BMP inhibition promotes the specification into paraxial mesoderm and skeletal muscles[2,18] whereas TGFβ inhibition further enhances the efficiency of hPSCs differentiation into somites[19]. Thus, we employed both BMP and TGFβ inhibition using the dual SMAD inhibitor cocktail (2SMADi)[19,20]. We first assessed the effect of 2SMADi at the NMP induction stage (day 0 – day 3) (Supplementary Fig. 2a). We instructed the generation of NMPs using our previously established protocol[21,22]. Initial exposure of hPSCs to WNT agonist (CHIR) and basic FGF (bFGF) signals for three days resulted in the efficient generation of NMP cells that co-expressed the nascent mesodermal marker BRACHYURY (TBXT) and the neural progenitor marker SOX2. Paraxial mesoderm TBX6+ cells were also evident mainly in the periphery of the NMP colonies (Supplementary Fig. 2b, c). We observed that simultaneous, early treatment of hPSCs with CHIR/bFGF and 2SMADi resulted in the generation of cells that expressed high levels of SOX2 and low levels of TBXT in the absence of TBX6 (Supplementary Fig. 2b, c, d). The low levels of TBXT agreed with the known role of SB431542 to mediate TBXT suppression[23]. Thus, the addition of 2SMADi changed the balance of TBXT and SOX2 towards a more SOX2-high fate (Supplementary Fig. 2e).

To determine the effect of 2SMADi in rostro-caudal identity, we analyzed the expression of the posterior markers *CDX2*, *NKX1.2*, and *HOXB1* and the anterior marker *OTX2* in hPSCs treated during day 0 – day 3 with (i) 2SMADi, (ii) WNT/bFGF/2SMADi and (iii) WNT/bFGF (Supplementary Fig. 2a). Cells treated with 2SMADi alone acquired an anterior neural identity expressing the neural progenitor markers *SOX2*, *SOX1* and *OTX2*, indicative of an anterior neural identity and in agreement with previously published studies[20] (Supplementary Fig. 2d). The WNT/bFGF condition alone resulted in the generation of NMPs that express the posterior markers *CDX2* and *NKX1.2* as previously described[21,22,24]. The simultaneous addition of 2SMADi and WNT/bFGF resulted in the generation of posterior cells expressing *CDX2* and *HOXB1*. A striking difference between the WNT/bFGF/2SMADi and WNT/bFGF treatments was the upregulation of *NKX1.2* and *SOX2* in the former (Supplementary Fig. 2d); this upregulation was associated with a pre-neural tube identity[25]. We next examined the potential of the WNT/bFGF/2SMADi treated cells to differentiate into neural and mesodermal progenitors. To enhance the generation of both cell types, we continued the treatment from day 3 to day 6 with WNT/bFGF/2SMADi in the presence of retinoic acid (RA) and the SHH smoothened agonist (SAG) to induce a ventral identity (Fig. 1a). Analysis of cells on day 6 revealed the exclusive differentiation into neural progenitor cells expressing *SOX2*, *SOX1*, *PAX6*, and *NKX1.2* (Fig. 1b, c). Genes associated with a presomitic (PSM) identity, such as *MEOX1*, *FOXC1*, *FOXC2*, and *MYF5* were not expressed (Fig. 1b). Thus, exposure of cells to the 2SMADi during the NMP induction stage precludes mesoderm formation. This was in agreement with the finding that 2SMADi promotes the preferential differentiation of hPSCs towards the neuroectodermal lineage[20,26].

We then explored whether applying 2SMADi after NMP induction could potentiate the simultaneous development and differentiation of neural and mesodermal progenitors (Fig. 1a). Indeed, exposure of NMP cells to WNT/bFGF/2SMADi in the presence of RA and SHH for three days (day 3 – day 6) resulted in the generation of clearly segregated neural and mesodermal progenitor clusters that self-organized in the dish (Fig. 1c). The neural progenitor clusters expressed SOX2, whereas the mesodermal clusters expressed PAX3 in the absence of SOX2 (Fig. 1d). We quantified the number of PAX3+ and SOX2+ cells in the H1, H9 and XM001 cell lines. All hPSC lines were able to generate the NP (SOX2+) and PSM (PAX3+) populations, confirming the reproducibility of the protocol (Fig. 1e). However, in agreement with our previous findings, we observed that the XM001 and H9 lines were more primed to differentiate towards the neural lineage[27]. Some PAX3+ cells were also evident in the neural SOX2+ clusters corresponding to dorsal neural progenitor identity. PSM-specific genes were also upregulated

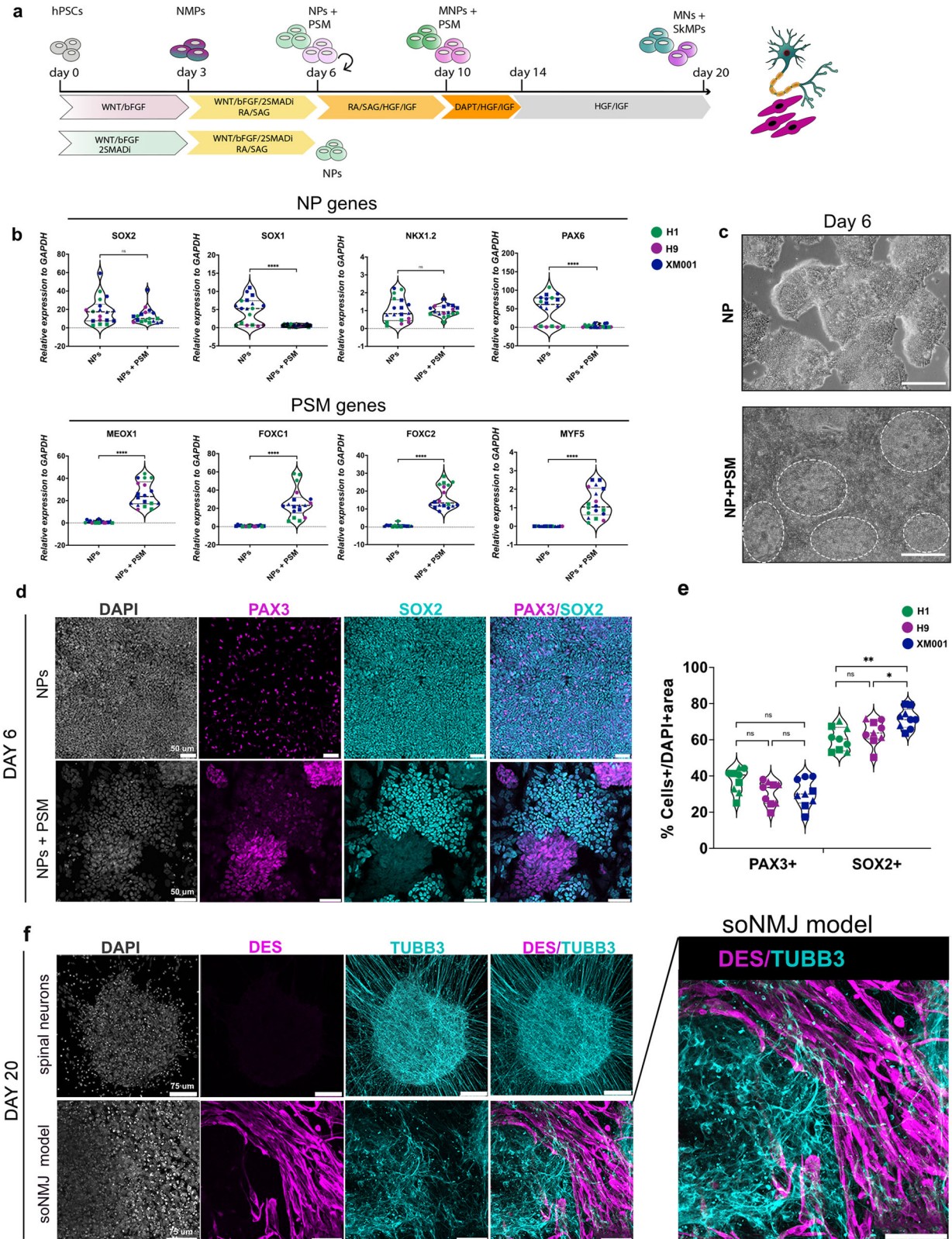

(Fig. 1b). Thus, we concluded that early treatment of hPSCs with 2SMADi, even in the presence of WNT/bFGF primes NMPs towards an exclusive neural identity, whereas treatment after the establishment of the NMP state, allows the simultaneous differentiation to both neural and PSM lineages at day 6.

We next sought to analyze the potential of these cultures to differentiate into spinal cord neurons and skeletal muscle cells.

Thus, we passaged the cells at day 6 and plated them in an optimized neurobasal (NB) medium containing 500 nM RA, 500 nM SAG, 2 ng/ml HGF and 2 ng/ml IGF, which could support both the specification of neural progenitor cells towards a ventral MN identity and the proliferation of mesodermal progenitors. RA and SAG were used to pattern the neuroectoderm towards a ventral MN identity at concentrations previously described by Maury et al.[22,25]. The inclusion of

**Fig. 1 | Generation of self-organizing neural and mesodermal progenitors from human PSCs in adherent culture. a** Schematic representation illustrating the strategy employed to generate the soNMJ model and NPs from hPSCs. **b** qPCR analysis on day 6 demonstrated that administration of dual SMADi (2SMADi) from day 0 – day 6 resulted in the differentiation of hPSCs to NPs expressing high levels of *SOX2, SOX1, NKX1.2* and *PAX6*. Administration of 2SMADi from day 3 – day 6 was sufficient to generate both NP cells and PSM cells expressing *MEOX1, FOXC1, FOXC2* and *MYF5*. The statistical tests employed included an unpaired t-test with Welch's correction. *$P \le 0.05$; **$P \le 0.01$; ***$P \le 0.001$; ****$P \le 0.0001$. Each dot with the same colour/shape represents different wells from a single differentiation experiment of a specific cell line. The dashed line is the median value. XM001: $N = 3$, $n = 9$; H1: $N = 2$, $n = 6$; H9: $N = 1$, $n = 3$. Source data are provided as a Source Data file. **c** Representative brightfield images of NP and NP + PSM cultures at day 6. Circles mark PSM colonies. Scale bars: 100 μm. **d** Immunofluorescence analysis at day 6 of differentiation showed that exposure to 2SMADi since day 0 resulted in the generation of SOX2⁺ NPs in the absence of mesodermal cells. Co-expression of PAX3 and SOX2 corresponded to dorsal NPs. Conversely, exposure to 2SMADi after day 3

resulted in the segregation of NMP cells to SOX2⁺ NPs and PAX3⁺ PSM cells. **e** We quantified the number of PAX3⁺ cells in H1 (37.5% ± 6.6%), H9 (30.6% ± 6.5%) and XM001 (30.6% ± 7.6) lines, and the number of SOX2⁺ cells in H1 (60.4% ±6.4%), H9 (64% ± 6.7%) and XM001 (71.8% ± 5.6%). The statistical tests employed included one-way ANOVA with Bonferroni's multiple comparison test. *$P \le 0.05$; **$P \le 0.01$; ***$P \le 0.001$; ****$P \le 0.0001$. Each dot with the same colour/shape represents a sample from a single differentiation experiment of a specific cell line. The dashed line is the median value. XM001: $N = 3$, $n = 9$; H1: $N = 3$, $n = 9$; H9: $N = 3$, $n = 9$. Scale bars: 50 μm. Source data are provided as a Source Data file. **f** Immunofluorescence analysis at day 20 of differentiation showed that early exposure to 2SMADi from day 0 - day 6 resulted in the generation of TUBB3⁺ neurons. Exposure to 2SMADi after day 3 resulted in the segregation of NMPs into DESMIN⁺ myoblast cells and TUBB3⁺ neurons. The immunofluorescence analysis was performed in the H1, H9 and XM001 cell lines (also see Table S1). Scale bars 75 μm. NPs Neural progenitors, PSM Pre-somitic mesoderm, 2SMADi dual-SMAD-inhibition, soNMJ model: self-organizing neuromuscular junction model.

HGF and IGF ensured the survival and proliferation of mesodermal progenitors[18]. At day 10, most neural cells expressed the MN progenitor marker OLIG2, at which timepoint progenitors were treated with the γ-secretase inhibitor DAPT to accelerate the generation of MNs[4]. Analysis on day 14 revealed the presence of both OLIG2⁺ cells and HB9⁺ differentiated MNs (Supplementary Fig. 3a). After day 14, cells were maintained in the presence of only HGF and IGF which were essential for the proliferation and survival of the muscle progenitor cells in the soNMJ model.

Comparison of the expression profile of genes associated with neural and mesodermal differentiation between day 3, day 6, and day 20 confirmed the differentiation of NMP cells to neural progenitors that expressed *SOX2, SOX1, OLIG2,* and more differentiated MNs that expressed *ISL1, FOXP1,* and *CHAT* (Supplementary Fig. 3b). As expected, on day 20, the expression of the muscle-specific progenitor markers *MYF5* and *MYOD* was upregulated. In contrast, the expression of the early mesodermal marker *MSGN1* was completely downregulated (Supplementary Fig. 3b). The early segregation of NP and PSM clusters, observed at day 6, was maintained and developed into clusters of differentiated neural TUBB3⁺ cells and muscle DESMIN⁺ progenitors at day 20 (Fig. 1f). Quantitative analysis of the neural and muscle-specific cell types revealed the presence of 18.9% ± 10.1% ISL1⁺ cells, 20.9% ± 7.6 % SOX1⁺ and 58.4% ± 10.7% MYF5⁺ cells (Fig. 2a, b).

Collectively, the data showed that 2SMADi during the NMP induction stage effectively eliminates the mesodermal lineage giving rise to posterior neurons. In contrast, delayed 2SMADi exposure, after the induction of NMP state, promotes the simultaneous development and differentiation of both neural and mesodermal lineages. Then, cells self-organize into distinct clusters of differentiated neurons and muscle progenitors. Thus, fine-tuning the timing of 2SMADi exposure allows the selective differentiation to either exclusively posterior neurons or neuromuscular cells.

### Efficient generation of a position-specific soNMJ model that includes supporting glial cells

Generating position-specific spinal cord neurons is necessary for proper disease modeling and transplantation studies. Thus, we next sought to characterize the rostocaudal and columnar identity of the spinal cord neurons in this neuromuscular model. It is well established that the identity of spinal cord neurons is instructed by the differential expression of HOX genes[28]. To define the rostrocaudal identity, we examined the expression of HOX proteins at day 20. Most MNs co-expressed HOXC6/ISL1 (80.2% ± 11.9%), confirming their predominant brachial spinal cord identity (Fig. 2c, e). We then assessed the acquisition of a defined columnar subtype identity. At the brachial spinal cord level, MNs are organized into two columnar identities corresponding to the lateral motor column (LMC) and median motor column (MMC) (Fig. 2d)[28]. Analysis with specific columnar identity

markers revealed that within the MN population, approximately 80% of the MNs co-expressed ISL1/LHX3 (72.4% ± 14.4%), associated with MMC identity while 20% co-expressed ISL1/2 /FOXP1 (25.3% ± 10%) associated with LMC identity (Fig. 2c, e).

Consistent with continuing maturation, at day 50, brachial spinal cord MNs were expressing the acetylcholine-synthesizing enzyme choline acetyltransferase (ChAT) and HOXC6 (Fig. 2f)[29]. It is worth noting that the MNs survived and matured in the neuromuscular model without external supplementation of neurotrophic factors (Supplementary Fig. 4a, b). Maturation of MNs in reductionist models depends on the addition of a medium enriched for glial-derived neurotrophic factor (GDNF), brain-derived neurotrophic factor (BDNF) as well as ciliary neurotrophic factor (CNTF)[4,29]. However, this is not required in this neuromuscular model, probably due to the presence of muscle and glial cells that support the MNs.

GFAP⁺ glia cells were first detected on day 50 without adding factors that support gliogenesis or serum[30]. This suggested that in the neuromuscular model neural progenitor cells can be maintained long enough to make the gliogenic switch. Indeed, the number of glial cells significantly increased from day 50 to day 100, recapitulating the later developmental timing of glial cells in vivo (Supplementary Fig. 4c, d). Additionally, day 50 immunostainings revealed the presence of V2a excitatory pre-motor interneurons, marked by expression of CHX10[31] and V1, V0, and dI6 interneurons marked by PAX2/LHX1 expression[32] (Supplementary Fig. 4e, f).

Therefore, timely application of the appropriate instructive and mitogenic signals until day 25 generates a neuromuscular model of defined rostrocaudal and columnar identity (Fig. 3a). Then, the co-developing cell types provide the necessary autocrine and paracrine signals giving rise to a dynamic, self-instructing model system that develops and matures with minimal requirements for exogenously added differentiation and maturation factors.

### High-content imaging reveals the self-organization of spinal cord neurons and skeletal muscles to form maturing neuromuscular junctions

Neuromuscular cultures were very dense by day 20 (Supplementary Fig. 5a), and we assessed whether cells would maintain their self-organizing capacity after passaging. Cultures were analyzed at day 50 using a high-content imaging system that allowed us to capture the whole culture dish. Immunofluorescence analysis with Myosin skeletal fast (Fast MyHC) and TUBB3 revealed the organization of skeletal myofibers in bundles surrounded by spinal cord neurons that could be maintained in culture for more than 100 days (Fig. 3a, b). 3D projection of the neuromuscular images revealed that, along the z-axis, the muscles and neurons were also segregating on different layers (Supplementary Movie 1), confirming that this is a self-organizing NMJ (soNMJ) model.

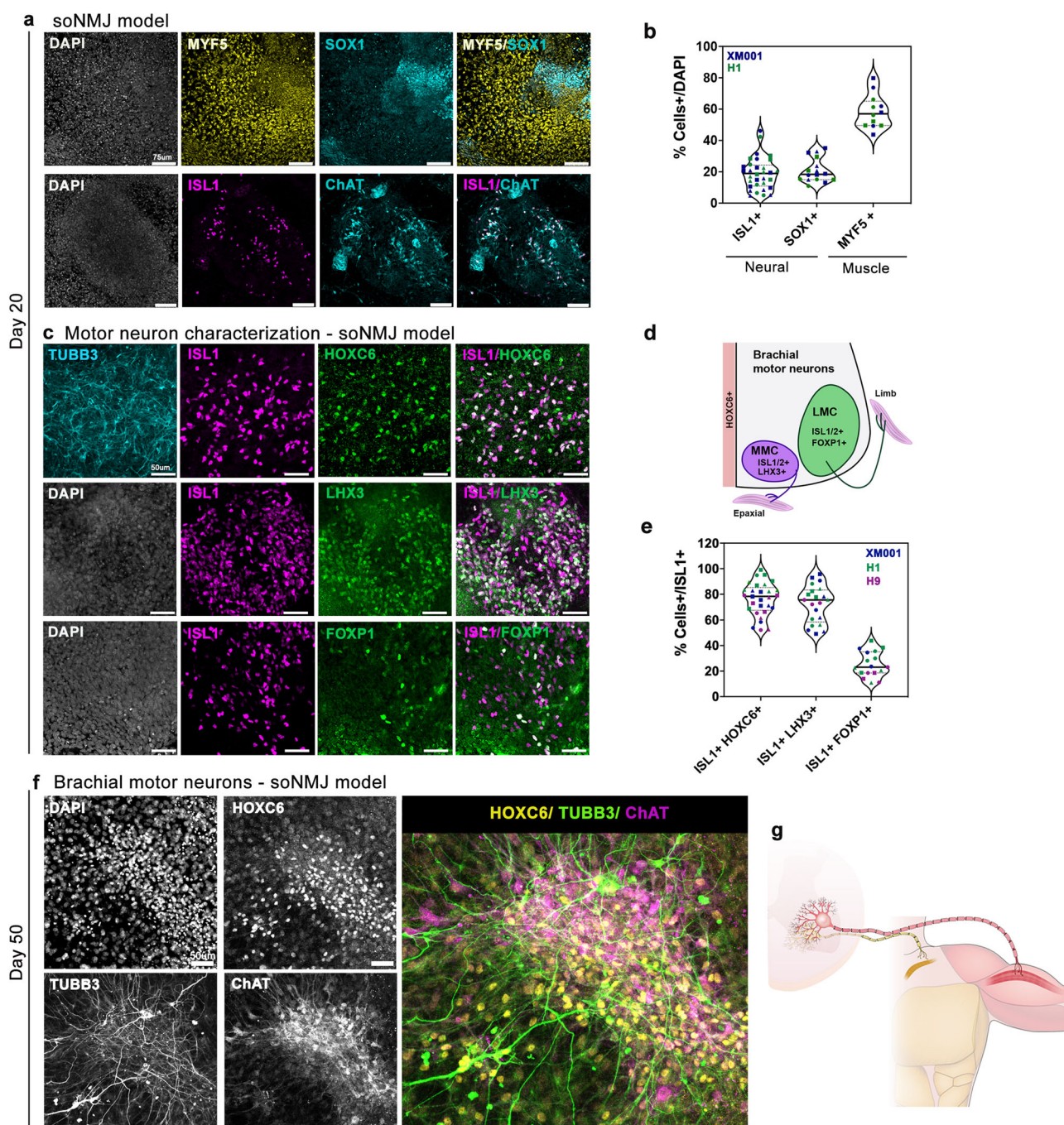

**Fig. 2 | Generation of position-specific brachial MNs corresponding to the MMC and LMC identity. a** Immunofluorescence analysis at day 20 in the soNMJ model revealed the presence of MYF5$^+$ skeletal muscle progenitors and SOX1$^+$ NPs. The presence of MNs was shown by the expression of ISL1 and ChAT. Scale bars: 75 µm. **b** Quantification of SOX1$^+$ NPs (20.9% ± 7.6%), MNs expressing ISL1 (18.9% ± 10.1%), and skeletal muscle progenitors expressing MYF5 (58.4% ± 10.7%). ISL1 (H1: $N = 3$, $n = 16$; XM001: $N = 3$; $n = 13$); SOX1 (H1: $N = 2$, $n = 6$; XM001: $N = 3$; $n = 9$); MYF5 (H1: $N = 2$, $n = 6$; XM001: $N = 2$; $n = 6$) Source data are provided as a Source Data file. **c** Analysis of the MN population at day 20 revealed the presence of brachial MNs co-expressing HOXC6 and ISL1. Both LMC MNs expressing ISL1/FOXP1 and MMC MNs expressing ISL1/LHX3 were present in the soNMJ model. Scale bars: 50 µm. **d** Schematic illustration of MN columnar organization at the brachial spinal cord.

**e** Quantification of HOXC6$^+$/ISL1$^+$ cells showing a prevalence of brachial MNs in the soNMJ model (80.2% ± 11.9%). Both MMC MNs (72.4% ± 14.4%), characterized by ISL1/LHX3 co-expression, and LMC MNs (25.3% ± 10%), characterized by ISL1/FOXP1 co-expression were present. ISL1 + HOXC6+ (H1: $N = 3$, n = 11; XM001: $N = 3$; $n = 9$; H9: $N = 3$; $n = 9$); ISL1 + LHX3+ (H1: $N = 3$, $n = 9$; XM001 $N = 3$; $n = 9$; H9: $N = 1$, $n = 3$); ISL1 + FOXP1+ (H1: $N = 3$, $n = 9$; XM001: $N = 1$; $n = 3$; H9: $N = 2$, $n = 4$). Source data are provided as a Source Data file. **f** Immunofluorescence analysis of brachial MNs expressing TUBB3, HOXC6, and ChAT at day 50. Scale bars: 50 µm. H1: $N = 3$, $n = 9$; XM001: $N = 3$, $n = 9$. **g** Schematic illustration of the neuromuscular connectivity in the human body at the forelimb level. soNMJ model: self-organizing neuromuscular junction model; NPs: neural progenitors; MNs: motor neurons; MMC: median motor column; LMC: lateral motor+ column.

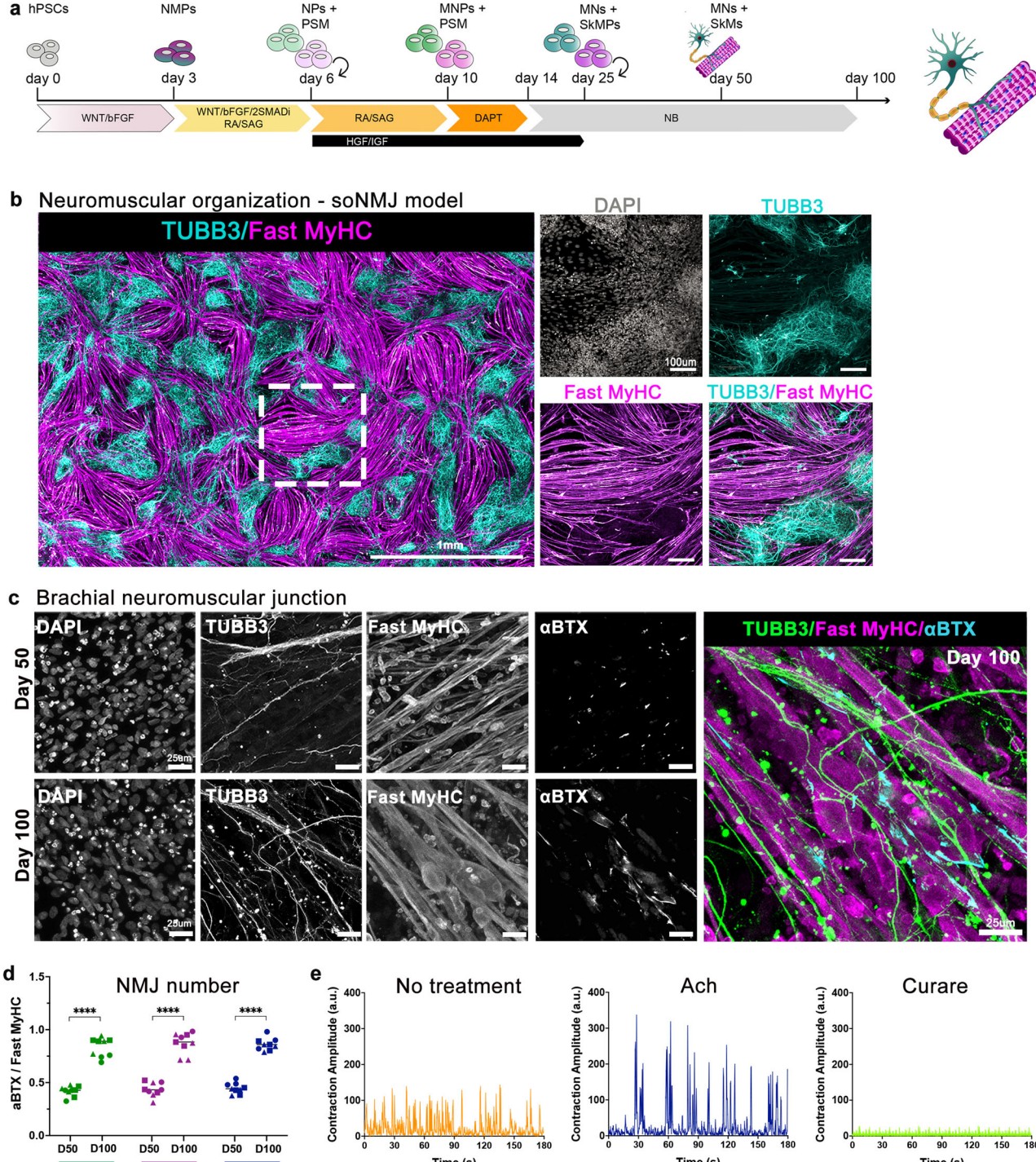

**Fig. 3 | Formation of functional NMJs in the soNMJ model. a** Schematic illustration of the strategy used to generate the soNMJ model from hPSCs under adherent culture conditions. **b** High-content imaging of whole well at day 50 soNMJ model. The representative image depicts the self-organization of the neurons TUBB3[+] (cyan) and skeletal muscle fibers that express Fast MyHC (magenta) in 20 fields acquired in the same well. Scale bars: 1 mm and 100 μm. H1: $N = 1$; H9: $N = 1$; XM001: $N = 1$. **c** Representative immunofluorescence image at day 50 and day 100 of differentiation showing the presence of TUBB3[+] neurites in contact with α-bungarotoxin[+] (αBTX[+]) AChR clusters on fast-twitch skeletal muscle fibers (Fast MyHC[+]) revealing the presence of neuromuscular junctions. Scale bars: 25 μm. **d** Quantification of the number of αBTX clusters at day 50 and day 100 of hPSC differentiation normalized to the number of Fast MyHC myofibers. The number of NMJs increased significantly from day 50 (H1: 0,419 ± 0,048; H9: 0,435 ± 0,067;

XM001: 0,447 ± 0,051) to day 100 (H1: 0,832 ± 0,09; H9: 0,872 ± 0,099; XM001: 0,861 ± 0,059). The statistical tests employed included unpaired t-test with Welch's correction. *$P ≤ 0.05$; **$P ≤ 0.01$; ***$P ≤ 0.001$; ****$P ≤ 0.0001$. Each dot with the same colour/shape represents the number of NMJs in a well from a single differentiation experiment of a specific cell line. The different colours correspond to different cell lines and the different shapes to different experiments (N). H1: $N = 3$, $n = 9$; H9: $N = 3$, $n = 9$; XM001: $N = 3$, $n = 9$. Source data are provided as a Source Data file. **e** At day 100 of differentiation, there was a spontaneous contraction of the skeletal muscles. Contraction analysis by live imaging showed that muscle contraction was increased by administering 10 μM acetylcholine and blocked by adding the acetylcholine receptor inhibitor, curare (10 μM). H1: $N = 3$, $n = 9$; XM001: $N = 9$, $n = 27$. soNMJ: self-organizing neuromuscular junction. Source data are provided as a Source Data file.

It was recently shown that skeletal myofibers self-organized in bundles instructed by mechanical tension in 2D cultures[33]. Still, in those cultures, the contraction of the skeletal muscles could not be maintained for too long due to loss of attachment. An important advantage of the soNMJ model is that muscles remained attached to the dish for long periods, even under contractile conditions (Supplementary Fig. 5a, c), probably due to the support that they received from the spinal cord neurons and glia cells. At day 50, we detected numerous acetylcholine receptor (AChR) clusters by staining for α-bungarotoxin (αBTX) contacted by TUBB3+ neurites, suggesting the formation of NMJs. To address the reproducibility of the soNMJ model, we analyzed the number of NMJs contacted by TUBB3+ neurites in three different PSC lines at day 50 and day 100 (Fig. 3c, d). The number of NMJs was similar among the different lines at day 50 when normalized to the total number of muscle fibers. In agreement with a stepwise, ongoing maturation, the number and size of the NMJs significantly increased from day 50 to day 100 (NMJs length >5 μm) (Fig. 3c, d and Supplementary Figure 5d). At day 100, the skeletal muscle fibers were multinucleated and had the typical striation of sarcomere organization (Supplementary Fig. 5b, Supplementary Fig. 6a, 6b). A characteristic of mature myofibers is the presence of PAX7+ muscle stem cells, also known as satellite cells. At day 100, we observed PAX7+ cells underneath the basal lamina-like area, an arrangement reminiscent of satellite cells in vivo[34] (Supplementary Fig. 5e and Supplementary Movie 6). Electron microscopy revealed the development of organized skeletal muscle exhibiting aligned sarcomeric units, mitochondria, t-tubules, and membrane folding with surrounding basal membrane and invagination sites with many caveolae (Supplementary Fig. 6a-e). The presence of synaptic vesicles in the presynaptic neuron further supported the formation of functional synapses (Supplementary Fig. 6f, g). Apart from MNs and skeletal muscles, terminal Schwann cells are necessary for the maturation and stability of NMJs[34]. Among the different cell types originating from NMPs, there are trunk neural crest cells[35] that generate terminal Schwann cells. Thus, we assessed the presence of terminal Schwann cells in the soNMJ culture using the expression of the S100β epitope at the NMJ. At day 50, S100β+ Schwann cells were already present and by day 100, they could be found at the NMJs (Supplementary Fig. 5f and Supplementary Movie 7).

Collectively, these data suggested that brachial MNs interacted with the skeletal myofibers to form NMJs supported by terminal Schwann cells. The increase in the number and size of the NMJs over time, as well as the presence of multinucleation and striation in the myofibers, suggested that the neuromuscular model matured over time. This was further supported by the detection of PAX7+ satellite-like cells and S100β+ terminal Schwann cells.

## Functional analysis of the soNMJ model

To investigate the functionality of the soNMJs, we analyzed the contractile muscle activity at day 75 after treatment with 10 μM acetylcholine and after treatment with 10 μM curare, which is a blocker of the AChR (Supplementary Movie 2). Treatment with acetylcholine increased the amplitude of muscle contraction. In contrast, treatment with curare blocked the contraction of the skeletal muscles (Fig. 3e). We further characterized the activity of the neuromuscular cultures using calcium imaging and optogenetic stimulation. To record the calcium activity, we used the ratiometric fluorescent calcium-binding dye Fluo8. Calcium transients were observed in spinal neurons and skeletal muscles that were distinguishable by speed and morphology (Fig. 4a and Supplementary Movie 3). Treatment with curare abolished the calcium activity in the skeletal muscles without affecting the firing of the neurons (Fig. 4a and Supplementary Movie 3).

To check the functionality of the spinal neurons, we performed patch-clamp recordings of the electrical activity. Neuronal somata were visualized by AAV-mediated neuron-specific expression of nucleus-targeted EGFP. Neurons had a resting membrane potential of −61.94 ± 5 mV (n = 5, corrected for liquid junction potential) and a low capacitance (18.72 ± 1.37 pF). Injection of 1-s long depolarizing currents elicited single or multiple action potentials (Fig. 4b), demonstrating functional neuronal physiology. Moreover, repetitive injections of short (10 ms) depolarizing currents at 10 Hz reliably elicited trains of action potentials (Fig. 4b). In ChR2-expressing neurons, we found that very brief light flashes could elicit similar repetitive firing (Fig. 4c). To obtain optogenetic control of neuronal firing, we transduced day 50 cultures with an adeno-associated virus (AAV) in which the expression of Channelrhodopsin2 (ChR2)-GFP is under the control of the human SYNAPSIN promoter. After 3-4 weeks, we performed whole-cell current clamp recordings on ChR2-GFP-expressing neurons, and found that trains of brief light flashes could elicit repetitive firing (Fig. 4c). The selective expression of ChR2 in spinal neurons allowed us to optogenetically stimulate synchronous neuronal firing, and thus to test for light-induced muscle contraction driven by synaptic transmission (Fig. 4d). At day 75, spinal neurons showed robust ChR2-GFP expression (Fig. 4e). Light stimulation of these cultures instructed the synchronous contraction of the skeletal muscles providing concrete evidence of controllable NMJ function. The addition of curare blocked the optogenetic stimulation of the skeletal muscles, further supporting our findings (Fig. 4f, Supplementary Movie 4).

The analysis revealed the presence of spinal cord neurons and skeletal muscles forming functional networks. This defined protocol takes advantage of the co-development of the neural and mesodermal lineages and their self-instructing and self-organizing attributes. The result is an efficient and robust platform for rapidly generating hPSC-derived functional soNMJs in 2D that could be used for disease modeling and high throughput studies.

## The soNMJ model as a platform to study spinal muscular atrophy

Spinal muscular atrophy (SMA) is a disease mostly caused by deletions in the SMN1 gene locus that lead to reduced levels of SMN protein[36]. While humans carry nearly identical copies of the SMN1 gene (SMN2) on the same chromosome, a base change in a splice site in exon 7 leads to insufficient expression of the full-length transcript[37]. SMN2 exists in a variable number of copies (2-8), which determines the severity of the disease[38]. SMA has been initially considered a MN-specific disease, but animal studies suggested an earlier effect at the NMJ and skeletal muscle[39–43]. Notably, the loss of the MN population usually only occurs at the end stages of the disease, and it might be a consequence of structural and functional defects at the NMJs that are the earliest pathological changes detected in SMA mouse models[44–47].

To provide a proof of concept for our soNMJ model, we used human induced pluripotent stem cells (hiPSCs) from two patients (SMA pt1, SMA pt2) with SMA to generate soNMJs (Fig. 5a). Both patients had a severe form of type 1 SMA with only two copies of SMN2. Due to the high presence of repetitive elements in the genomic locus of SMN1, it was not possible to target the deleted sequence to generate an isogenic line and thus we compared them to well-characterized control PSC lines. Both patient iPSC lines were first analyzed for the expression levels of SMN protein by immunofluorescence and western blotting, which revealed a reduction in SMN protein levels compared to the control (Supplementary Fig. 7a, 7b).

We further used the SMA lines to generate soNMJs in vitro through an NMP intermediate state (Supplementary Fig. 7c, 7d). The SMA neuromuscular cultures were analyzed at the progenitor (day 20) and mature state (day 50) for the presence of neural and muscle-specific progenitor cells and the formation of NMJs (Supplementary Fig. 7e and Fig. 5b). At day 20, DESMIN+ myoblast cells and SOX1+ NPs were detected in both the SMA type 1 and control hPSC lines (Supplementary Fig. 7e). However, at this stage, SMA myofibers were already smaller and less elongated in the SMA type 1 derived soNMJs

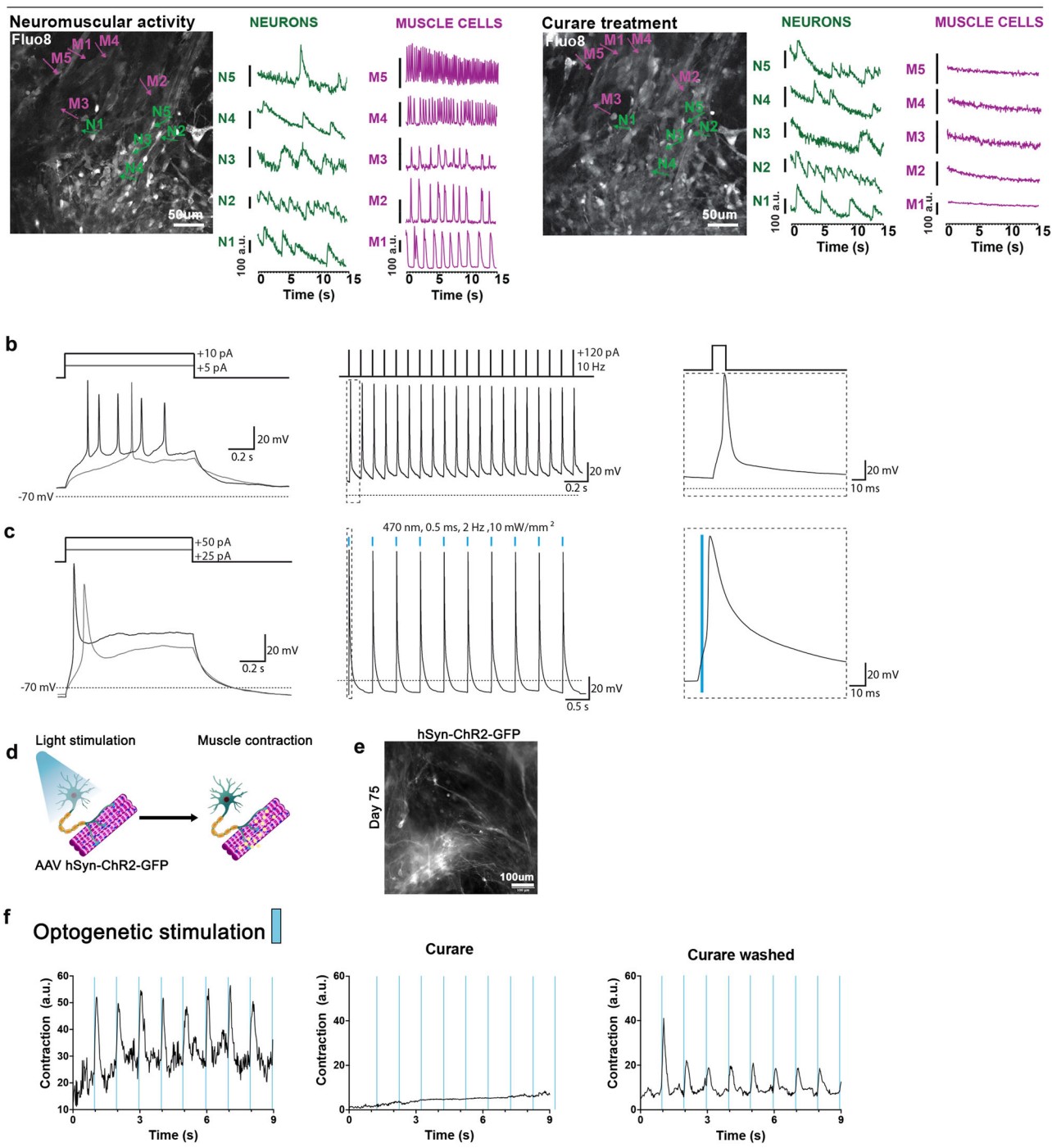

(Supplementary Fig. 7e). This suggested a delay in the differentiation and maturation of the skeletal muscle cells. At day 20, ChAT[+] MNs were also present in both SMA and control cultures but without detectable morphological differences (Supplementary Fig. 7f). This suggested that the MNs were not affected at this point.

Next, we analyzed the cells at day 50 for the presence of mature Fast MyHC[+] skeletal myofibers and TUBB3[+] neurons (Fig. 5b). At this stage, the SMA muscle cells were smaller and showed reduced elongation and organization (Fig. 5b). Additionally, at day 50, SMA myoblasts generated Fast MyHC[+] myofibers that were reduced in length, disorganized, pointing at different directions instead of aligning, and failing to form bundles (Fig. 5d). These results strongly suggested an

impaired maturation of the skeletal muscle cells in SMA. We further performed high-throughput analysis of the SMA and control line in 96-well formats and quantified the total muscle (Fast MyHC) and neural area (TUBB3) (Supplementary Fig. 8a). The analysis confirmed the significant reduction in the muscle area while we could not observe an effect in the neural area at this stage (Supplementary Fig. 8b, 8c). Detailed analysis of the MN area with the neurofilament marker SMI32 revealed that there was no significant difference between the SMA and control line (Supplementary Fig. 8d, 8e).

Synaptic defects at the NMJs are one of the hallmarks of SMA pathology[44]. We analyzed the presence of AChR clusters to investigate whether this phenotype was reproduced in the SMA soNMJ model.

**Fig. 4 | Functional characterization of the soNMJ model. a** The neuromuscular cultures were incubated with Fluo8-AM at day 75 for visualizing calcium transients with a spinning disk confocal microscope. Representative frames are shown for both spontaneous and 10 μM curare conditions. Individual neurons and muscle cells were identified by their morphology, and their calcium transients were plotted separately. Administration of 10 μM curare blocked the calcium activity in the skeletal muscle fibers but not in the neurons, supporting that the muscle contraction was driven by the neurons through the NMJs. Scale bars: 50 μm. H1: $N = 2$; XM001: $N = 1$. Source data are provided as a Source Data file. **b** Current clamp recording from a neuron transduced with an AAV encoding hSYN:NLS-GFP to label neuronal nuclei. The static current injection caused spontaneous action potential firing, while 10 Hz stimulation with brief square pulses revealed the potential for the repetitive firing of the neurons. Inset shows the first action potential. **c** Current clamp recording from a neuron transduced with an AAV encoding hSYN:ChR2(H134R)-GFP. The neuron reliably fired action potentials in response to 0.5 ms blue light (470 nm) flashes delivered at 2 Hz. Inset shows the first light-evoked action potential. **d** schematic representation of the optogenetic experiments. Neurons transduced with an AAV encoding hSYN:ChR2(H134R)-GFP were stimulated with blue light to drive the contraction of the skeletal muscle cells. **e** Immunofluorescence image of day 75 soNMJ culture model showing neurons expressing ChR2-GFP 3 weeks post-transduction with an AAV encoding hSYN:ChR2(H134R)-GFP. Scale bars: 100 μm. H1: $N = 1$; XM001: $N = 2$. Source data are provided as a Source Data file. **f** Optogenetic analysis of the soNMJ model at day 75. Every stimulation of ChR2-GFP$^+$ neurons by 470 nm light pulse resulted in the concomitant contraction of the skeletal muscle cells. Exposure to 10 μM curare blocked the neural transmission at the NMJs resulting in the inhibition of muscle contraction even upon optogenetic stimulation. The muscle contraction response to blue light stimulation of the neurons was restored by washing out curare.

Quantification of the αBTX$^+$ clusters showed that, in SMA, NMJs were significantly reduced in number and size compared to the hPSC lines H9, H1 and XM001 (Fig. 5b, c). Analysis of muscle contraction showed the presence of short-duration weak movements in individual muscle fibers, followed by exhaustion and a complete lack of activity. Thus, the contractility of the SMA skeletal myofibers was severely impaired (Fig. 5e, Supplementary Movie 5). We subsequently tested if the addition of acetylcholine would stimulate the muscle contraction and whether the administration of curare would block the spontaneous residual muscle contractility. Acetylcholine administration did not stimulate contraction, whereas curare failed to stop spontaneous muscle activity, suggesting the lack of functional NMJs in the SMA model (Fig. 5f, Supplementary Movie 5). Light stimulation at day 50 failed to instruct the synchronous contraction of the skeletal muscles providing concrete evidence of a functional defect (Supplementary Fig. 8f, Supplementary Movie 8).

Collectively, these data revealed an early muscle-specific phenotype and an apparent impairment in the formation of synapses between the motor neuron endplates and the skeletal muscle in SMA prior to motor neuron cell death. Thus, the soNMJ model provides unique opportunities for mechanistic studies that will elucidate the sequence of events leading to neuromuscular diseases in humans.

## Discussion

Here, we present a bottom-up approach to generate functional self-organizing NMJs in vitro from hPSCs. We show that the combinatorial and temporal application of small molecules supports the initial patterning of neural and mesodermal cell types from hPSCs through an NMP state. Consistent with previous studies, early exposure of hPSCs to 2SMADi restricts the potential of NMP cells to generate mesoderm and instructs them toward an exclusively neural spinal cord identity[5,26]. Strikingly, exposure to 2SMADi after the NMP state supported the growth and differentiation of spinal cord neurons and skeletal muscles that self-organized to generate functional soNMJs. Thus, fine-tuning the exposure time to 2SMADi is sufficient to generate either spinal neurons or a complete brachial soNMJ model from hPSCs.

A significant advantage of the soNMJ model compared to previously published NMJ models[2,8,9,11,12,14,48] is that we can observe the co-development and clear segregation of the neural and mesodermal progenitors at an early stage, corresponding to day 6 of differentiation. Passaging the cells at that stage is crucial in order to retain the mesodermal cells that form dome-like structures prone to detachment surrounded by neural progenitor cells. Thus, day 6 of differentiation emerges as an early quality control checkpoint for the robust generation of complex neuromuscular cultures through a neuromesodermal progenitor stage. It is essential to emphasize that these cultures are very rich, encompassing a large number of different cell types such as spinal cord motor neurons, interneurons, glia cells, terminal Schwann cells, skeletal muscles and satellite cells. Thus, these cultures more closely resemble the diverse composition observed in vivo as compared to simple monocultures or co-cultures of solely two different cell types.

Another important advantage of the soNMJ model over co-culture models is that it can be maintained for more than 100 days and that NMJs form and mature during this time in the absence of exogenous stimuli. This supports the notion that co-developing cell types provide the necessary support that gives rise to a dynamic, self-instructing model system that develops and matures with minimal requirements for exogenously added differentiation and maturation factors. To our knowledge, this is the first position-specific NMJ model that can self-organize and be maintained for extended periods of time under adherent cell culture conditions. Furthermore, the contraction of the skeletal muscles, in the soNMJ model, is instructed by the MNs and persists for long periods of time without the detachment issues that affect other models[11,49]. This is probably due to the continuous support that the muscle cells receive from the neurons and glia present in the soNMJ model.

Analysis of the rostrocaudal identity revealed the generation of predominantly brachial MNs corresponding to both median and lateral motor columns. This demonstrates the feasibility of generating position-specific NMJ models from hPSCs. Further studies will be needed to enable the generation of soNMJs corresponding to the thoracic and lumbar spinal cord levels through the timely application of appropriate posteriorizing signals. The development of position-specific NMJ models will open up an exciting opportunity to study the selective vulnerability of position-specific neurons to neuromuscular diseases.

The establishment of the soNMJ model offers an efficient, defined, and robust platform that can be used for mechanistic studies and high throughput approaches. The simplicity and scalability of the soNMJ model are ideal for drug screening studies and the development of novel therapeutic approaches for neuromuscular diseases, paving the road to personalized medicine.

## Methods

### Human pluripotent stem cell lines and culture conditions

The female H9 (WiCell, WA09) and male H1 (WiCell, WA01) hPSC lines were approved for use in this project by the regulatory authority for the import and use of human embryonic stem cells in the Robert Koch Institute (AZ:3.04.02/0123) and the female XM001 iPSC line (HMGUi001-A)[50] were cultured in mTESR1 medium (Stem Cell Technologies) on Geltrex LDEV-Free hESC-qualified reduced growth factor basement membrane matrix (Life Technologies) at 37 °C and 5% $CO_2$. We obtained the SMA patient 1 line (SMA E1C4)[51] from Emory University and the SMA patient line 2 (CS32iSMA-nxx) from Cedars-Sinai which are both males. The SMAE1C4 cell line was generated from fibroblasts that were reprogrammed using the CytoTune-iPS 2.0 Sendai virus reprogramming kit (ThermoFisher), following the manufacturer's protocol. Briefly, fibroblasts were transduced with a Sendai virus cocktail containing four transcription factors (OCT3/4, SOX2,

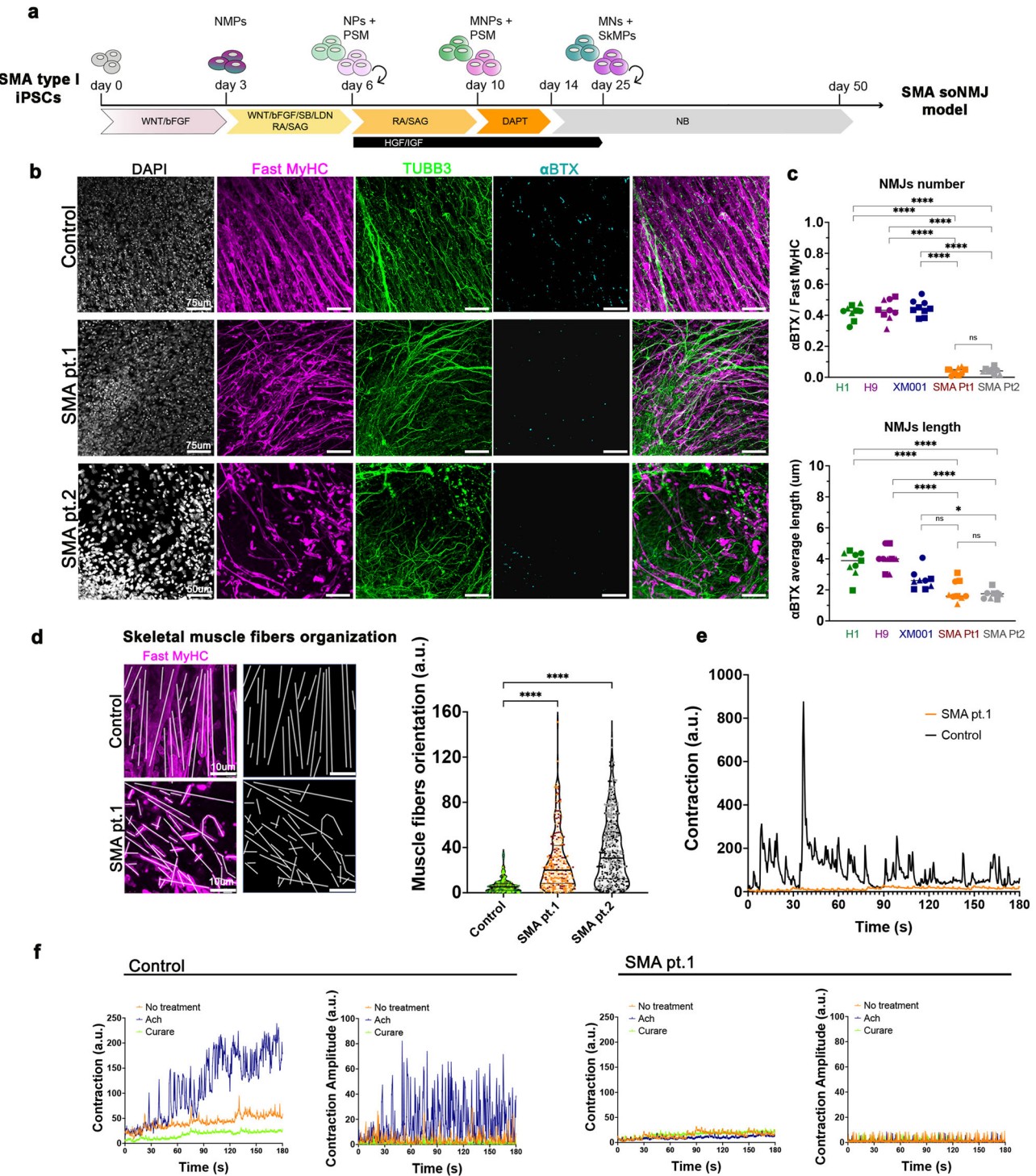

**Fig. 5 | Generation of an all human SMA soNMJ model. a** Schematic illustration of the soNMJ differentiation from SMA type I hiPSCs. **b** Representative images of αBTX clusters in SMA iPSC derived soNMJs and control at day 50 of differentiation. Scale bars: 75 μm. **c** Quantifications at day 50 of the αBTX⁺ clusters revealed a significant reduction in the number and size of αBTX clusters in the SMA type I soNMJ cultures compared to H1, H9, and XM001 control lines. The statistical tests employed included one-way ANOVA with Bonferroni's multiple comparison test. *$P \leq 0.05$; **$P \leq 0.01$; ***$P \leq 0.001$; ****$P \leq 0.0001$. Each dot with the same colour/shape represents the number of NMJs in a well from a single differentiation experiment of a specific cell line. The different colours correspond to different cell lines and the different shapes to different experiments (N). H1: $N = 3$, $n = 9$; H9: $N = 3$, $n = 9$; XM001: $N = 3$, $n = 9$; SMA pt.1: $N = 3$, $n = 9$; SMA pt.2: $N = 3$, $n = 9$.

**d** Immunofluorescence analysis using Fast MyHC marker at day 50 showed compromised skeletal muscle fiber organization in the two SMA type I hiPSC lines compared to H1 hPSC control. Results are shown as the angle of deviation of each individual fiber from the average fiber orientation. H1: $N = 3$, $n = 9$; SMApt1: $N = 3$, $n = 9$; SMApt2 $N = 3$, $n = 9$. Scale bars: 10 μm. **e** Skeletal muscle contraction was impaired in SMA type I compared to XM001 control hiPSC line at day 50 of differentiation. H1: $N = 3$, $n = 9$; XM001: $N = 3$, $n = 9$; SMA pt1: $N = 3$, $n = 9$. **f** Muscle contraction analyzed in SMA type I at day 50 did not respond to 10 μM acetylcholine and 10 μM curare administration compared to H1 hPSC control line. hiPSCs: Human induced pluripotent stem cells; soNMJ: self-organizing neuromuscular junction.

KLF4, c-MYC) and maintained in fibroblast medium (DMEM, Gluta-MAX, FBS; ThermoFisher) for one week. Transduced fibroblasts were replated onto Vitronectin-coated dishes in mTESR media (StemCell-Technologies) and individual iPSC colonies began to emerge within two weeks. We manually isolated clonal iPSC colonies and expanded two clones for characterization using pluripotency markers (NANOG, OCT3-4, TRA1-60, SSEA4) and the StemMAC trilineage differentiation kit (MACS Miltenyi Biotec). All the patient cell lines and protocols were approved by review boards at Emory University and Cedars-Sinai Medical Center. Cell lines were checked for normal karyotype and were mycoplasma free. The cells were passaged twice a week using versene solution (Thermo Fisher).

### In vitro generation of NMPs from human pluripotent stem cells
Human PSCs were grown for at least two passages and after they reached 70% confluency, they were dissociated into single cells using Accutase (Life Technologies). Single cells were counted using the Countess II automated cell counter (Thermo Fisher) and were plated on 35 mm cell culture dishes (Corning) coated with Matrigel (Life Technologies) at a density of $75.000 - 125.000 / cm^2$. The initial plating concentration of the PSCs was adjusted depending on the growth rate of the specific human pluripotent stem cell line (XM001: $75.000 / cm^2$; H9 line: $110.000 / cm^2$; H1 line: $100.000 / cm^2$).

For NMPs differentiation, Human PSCs were plated in neurobasal (NB) medium supplemented with $10 \mu M$ Rock inhibitor (Tocris Bioscience), $3 \mu M$ CHIR99021 (Tocris Bioscience) and 40 ng/ml bFGF (Peprotech). NB is a 1:1 medium of Advanced Dulbecco's Modified Eagle Medium F12 supplemented with 1 x N2 (GIBCO), and Neurobasal medium (GIBCO) supplemented with 1 x B27 (GIBCO), 2 mM L-glutamine (GIBCO), 75 μg/ml BSA fraction V (Sigma), 0.1 mM 2-mercaptoethanol (GIBCO). The next day ROCK inhibitor was removed and the cells were maintained in NB medium supplemented with $3 \mu M$ CHIR99021 and 40 ng/ml bFGF (Peprotech) until day 3. In the SMA Pt2 line, 10 ng/ml BMP4 were added from day 1- day 2 to support the efficient generation of NMPs due to the low endogenous BMP levels. The medium was changed daily. On day 3 the cells were analyzed by immunofluorescence for the co-expression of the NMP markers T/BRA and SOX2.

To test the effect of 2SMADi on NMPs, the cells were plated in neurobasal (NB) medium supplemented with $10 \mu M$ Rock inhibitor (Tocris Bioscience), $3 \mu M$ CHIR99021 (Tocris Bioscience), and 40 ng/ml bFGF (Peprotech), $10 \mu M$ SB431542 (Biozol) and 0.2 nM LDN193189 (MedChem). The next day Rock inhibitor was removed and the cells were maintained in NB medium supplemented with $3 \mu M$ CHIR99021, 40 ng/ml bFGF (Peprotech), $10 \mu M$ SB431542 (Biozol) and 0.2 nM LDN193189 (MedChem) until day 3. The medium was changed daily.

### In vitro generation of soNMJ model
At day 3 of NMPs differentiation, the medium was changed to NB supplemented with $3 \mu M$ CHIR99021 (Tocris Bioscience) and 40 ng/ml bFGF (Peprotech), $10 \mu M$ SB431542 (Biozol) and 0.2 nM LDN193189 (MedChem), 500 nM RA (Sigma) and 500 nM SAG (Calbiochem Millipore). The medium was changed daily until day 6. At day 6, a single-cell suspension was obtained by incubating the cells with Accutase (Life Technologies) for 3 minutes at 37 °C. A density of 100.000 cells/cm² was seeded into 35 mm plates (IBIDI) coated with matrigel (Life Technologies). The cells were plated in NB medium with $10 \mu M$ Rho-associated protein kinase ROCK inhibitor (Tocris Bioscience), 2 ng/ml IGF1, 2 ng/ml HGF (Peprotech), 500 nM RA (Sigma) and 500 nM SAG (Calbiochem Millipore). The following day ROCK inhibitor was removed and the cells were maintained in NB medium supplemented with 2 ng/ml IGF1, 2 ng/ml HGF (Peprotech), 500 nM RA (Sigma) and 500 nM SAG (Calbiochem Millipore). The medium was changed daily until day 10. At day 10, the medium was replaced with NB supplemented with $10 \mu M$ DAPT, 2 ng/ml IGF1, and 2 ng/ml HGF (Peprotech)

until day 14. The medium was changed daily until day 14. At day 14, DAPT was removed and the cells were kept in NB medium with 2 ng/ml IGF1 and 2 ng/ml HGF (Peprotech). The medium was changed every other day until day 25. At day 25, the cells were dissociated into clumps using TrypLE Express (Life Technologies) for 3 minutes at 37 °C and replated from the 35 mm plates (IBIDI) into 12-well-plates with 18-20 mm glass coverslips coated with matrigel (Life Technologies). The clumps were plated with a dilution of 1:6 using NB medium supplemented with $10 \mu M$ Rho-associated protein kinase ROCK inhibitor (Tocris Bioscience), 2 ng/ml IGF1, 2 ng/ml HGF (Peprotech) and 1x ITS (Life Technologies). The cells were kept in this medium for three days (until day 28) and the medium was changed daily. After day 28 the cells were maintained in NB medium supplemented with 1x ITS (Life Technologies), and the medium was changed every other day. The formation of neuromuscular junctions was analysed between day 50 and day 100 by fluorescence immunostaining.

### In vitro generation of brachial motor neurons
At day 3 of npNMPs differentiation, the medium was changed to NB supplemented with $3 \mu M$ CHIR99021 (Tocris Bioscience) and 40 ng/ml bFGF (Peprotech), $10 \mu M$ SB431542 (Biozol) and 0.2 nM LDN193189 (MedChem), 500 nM RA (Sigma) and 500 nM SAG (Calbiochem Millipore). The medium was changed daily until day 6.

At day 6, a single cell suspension was obtained by incubating the cells first with Versene (Life Tecnhologies) for 3 minutes at 37 °C. Afterwards, Versene was removed and the cells were incubated with Accutase (Life Technologies) for 3 minutes at 37 °C. A density of 100.000 cells/cm² was seeded into 35 mm plates (Corning) coated with Matrigel (Life Technologies). The cells were plated in NB medium with $10 \mu M$ Rho-associated protein kinase ROCK inhibitor (Tocris Bioscience), 500 nM RA (Sigma) and 500 nM SAG (Calbiochem Millipore). The following day ROCK inhibitor was removed and the cells were maintained in NB medium supplemented with 500 nM RA (Sigma) and 500 nM SAG (Calbiochem Millipore). The medium was changed daily until day 10. At day 10, the medium was replaced with NB supplemented with $10 \mu M$ DAPT until day 14. At day 14, DAPT was removed and the cells were kept in NB medium for a maximum of 50 days.

### Fluorescence immunostainings analysis
The cells were washed once with PBS and fixed with 4% PFA for 20 minutes at 4 °C. Then PFA was removed and the cells were washed three times with PBS with 0.3% Triton X-100 (Sigma Aldrich) and blocked in PBS with 4% Bovine Serum Albumin (BSA) (Sigma Aldrich) and 0.3% Triton X-100 (Sigma Aldrich) for 1 hour at room temperature. Alternatively, the cells could be maintained at 4 °C in blocking solution for up to two weeks.

After blocking, the cells were incubated with primary antibodies overnight at 4 °C (Table S2). Primary antibodies were washed three times with PBS with 0.3% Triton X-100 for 5 min and then the cells were incubated with secondary antibodies for 1-2 hours at room temperature (Table S2). After the incubation with the secondary antibody, the cells were washed three times with PBS with 0.3% Triton X-100 for 5 min and incubated with DAPI in PBS. After the incubation with DAPI, an additional washing with PBS was performed before mounting the coverslips on glass slides. Fluorescent images were acquired using a Leica SP8 confocal microscope.

### High-content image analysis
High-content automated confocal imaging of the neuromuscular cultures was performed using an Opera Phenix™ High-Content Screening System (Phenix; Perkin Elmer) and Harmony® v5.1 (Perkin Elmer) software. To this end, the cells were re-plated at day 25 on 96-well plates (PerkinElmer), cultured until day 50 and then fixed and stained as described above.

Images were captured using a 20x water immersion objective with 49 fields-of-view per well of a 96-well plate (PhenoPlate™; Perkin Elmer) with z-Stack size of 88 μm and a distance of 2 μm. Image analysis was performed in Harmony Software (Perkin Elmer) to evaluate the muscle, neural and motor neuron area. The muscle, neural and motor neuron area was normalized to the DAPI⁺ area and is indicated in percentage as a fraction of the total cell culture area. These values were obtained from three independent experiments ($N = 3$) and compared using an unpaired t-test with Welch´s correction to determine statistical differences ($p < 0.05$) using Graph-Pad Prism 9.

### Reverse transcription – quantitative PCR Analysis
Total RNA was isolated from cells growing on matrigel using the RNeasy kit (QIAGEN) according to the manufacturer's instructions and digested with DNase I (QIAGEN) to remove genomic DNA. First strand cDNA synthesis was performed with Superscript III system (Invitrogen) using random primers and amplified using Platinum SYBR-Green (Invitrogen). For qPCR the Applied Biosystems Quantstudio 6 Flex Real-Time PCR system was used. PCR primers were designed using NCBI Primer-Blast software, using exon-spanning junctions (Table S3). Expression values for each gene were normalized against GAPDH, using the delta CT method and standard deviations were calculated and plotted using GraphPad Prism 9. To compare gene expression data at different time points, the mean expression value at day 3 was calculated and used to determine the fold change for each gene at different time points. Error bars represent standard deviation across three biological replicate samples.

### Western blot
Whole protein extraction was performed with RIPA buffer (ThermoFisher) and Protease inhibitor (Peprotech) and proteins were quantified using the Pierce™ BCA Protein Assay Kit (Thermo Fisher Scientific) according to manufacturer's instructions in a 96-well format. 20 μg of protein were loaded on 10% SDS-page gel and run at 120 V for 1 h. Semi Dry transfer to PVDF membranes was performed by using the Trans-Blot Turbo Transfer System (BioRad) for 30 min at 35 V. After that, the membranes were blocked with 3% skimmed milk in TBS with 0.1% Tween20 for 1 h at room temperature. After 1 h blocking at room temperature, the membrane was incubated with the SMN antibody (1:10,000 in blocking solution) or Beta-actin antibody (1:20,000 in blocking solution) over night at 4 °C under continuous shaking. The next day, the membrane was washed 3 times with TBS with 0.1% Tween20 (washing solution) for 5 min under continuous shaking. Then, the membrane was incubated with the secondary antibody HRP anti-mouse (Sigma; 1:10,000 in blocking solution) for 2 h at room temperature under continuous shaking. The membrane was washed 3 times with washing solution for 5 min. Finally, the protein bands were detected using the ECL™ Start Western Blotting Detection Reagent according to manufacturer's instructions and the Chemidoc MP Imaging System (Bio-Rad). All antibodies used are listed in Table S2.

### Transmission electron microscopy
The soNMJ cultures were fixed at day 75 with 2% (w/v) paraformaldehyde and 2.5 % (v/v) glutaraldehyde in 0.1 M phosphate buffer for 2 hours at room temperature. Samples were postfixed with 1% (v/v) tannic acid, 1% (v/v) osmium tetroxide, 0.5% (v/v) uranyl acetate, dehydrated in a graded series of ethanol, and embedded in PolyBed® 812 resin (Polysciences, Germany). Ultrathin sections (60-80 nm) were poststained with lead citrate, and examined at 80 kV with a Zeiss EM 910 transmission electron microscope (Zeiss, Germany). Acquisition was done with the Quemesa CDD camera using the iTEM software (Emsis GmbH, Germany).

### Calcium imaging with Fluo8
Between day 50 and day 100, the cells were incubated at 37 °C with 4 μM of the cell-permeant calcium indicator Fluo-8 AM (Abcam). After 30 minutes of incubation, the cells were washed two times with culture medium (NB without phenol red) and left to recover for 15 min in the incubator prior to the start of any optical recording. During acquisition, the samples were kept at 37 °C with 5% CO2. Fluorescent time series images were acquired with a CSU-W1 spinning disk confocal microscope. The calcium transients were recorded for 2–3 minutes at 20 frames per second in three different conditions: without any treatment, with the addition of 10 μM Acetylcholine (Sigma Aldrich) and with addition of 10 μM Curare (Sigma Aldrich).

Analysis of the calcium activity was performed by measuring the variation of fluorescence intensity in each ROI with ImageJ (Fiji version 2.14). ROIs were detected manually, and neurons and muscle cells could be distinguished by morphology. The variation of fluorescence intensity for each individual ROI was then plotted in Prism 9 (GraphPad).

### Electrophysiology
In order to visualize neuronal somata for targeted patch clamp recordings, cultures were incubated 48 h with 1E + 9 VG/mL AAV9 expressing EGFP under the neuron-specific hSYN promoter 10 days before the experiments. Recordings were performed on an inverted microscope (IX73 Olympus) at room temperature, using a Multiclamp 700B amplifier under control of a Digidata 1550 digitizer board and Clampex 10 software (Molecular Devices), which was also used to trigger the LED. Fluorescent light from an LED (pE-4000, CoolLED) was passed through an appropriate filter (DAPI/FITC/Cy3/Cy5 Quad sbx LED HC Filter Set, AHF) and a 20x dry objective (UPLSAPO20X/0.75 U Plan S Apo). Extracellular solution contained (in mM): 140 NaCl, 2.4 KCl, 10 HEPES, 10 glucose, 2 CaCl2, 1 MgCl2 (pH 7.3, 300 mOsm); intracellular solution contained (in mM): 135 CH3KO3S, 4 NaCl, 2 MgCl2, 10 HEPES, 2 ATP-2Na, 0.3 GTP-Na, 0.06 EGTA, 0.01 CaCl2. Whole-cell current-clamp recordings were performed with bridge balance compensation, and membrane voltage was adjusted manually to <−60 mV if necessary.

### Channelrhodopsin-stimulation
For specifically stimulating neurons in the neuromuscular culture, cells were incubated with 1E + 9 VG/mL AAV8-hSyn-hChR2(H134R)-GFP for 48 h at day 50. After 3-4 weeks, the expression of ChR2(H134R)-GFP was assessed using a confocal microscope. Cultures were transferred to the electrophysiology setup described above, and continuously superfused with HEPES-buffered extracellular solution at room temperature. Bright-field images of 2048 × 2048 were acquired at 10 Hz using an sCMOS camera (Photometrics Prime BSI Express) with 50 ms exposure time. Every 50 s, ChR2-expressing neurons were optically stimulated with three 5-ms flashes of 470 nm light at 10 Hz, with 10 mW/mm² intensity.

For the analysis, the muscle contraction recorded in brightfield was quantified using MuscleMotion (described below), plotted in GraphPad Prism 9 and correlated to the light pulses.

### Muscle fiber orientation analysis
The orientation of the skeletal muscle fibers was analyzed in control and SMA soNMJ models at day 50. For this, the angle of individual skeletal myofibers was measured in each confocal image. Fast myosin-heavy chain was used as a marker for fiber identification. The resulting measurements from each image were then averaged and the deviation of each measurement from the average was then calculated and plotted in GraphPad Prism 9. The higher the value, the higher the difference between a single fiber angle and the average value, and the higher the variability of myofibers orientation. Three different images were

analyzed in each experiment and three independent experiments were run for each cell line.

## Contraction analysis of soNMJ model

Three different regions of the same coverslip were recorded for contraction analysis using a Leica SP8 confocal microscope. Each region was recorded twice for 3 minutes using brightfield with 20x zoom. To quantify the muscle contraction, each movie was analyzed individually using MuscleMotion (an open-source ImageJ Macro). A reference frame was automatically detected by MuscleMotion to measure the contraction as a variation in pixel over time. The results were then plotted in GraphPad Prism 9.

## Quantifications and statistical analysis

Quantifications of day 3 NPs, NMPs and 2SMADi-treated NMPs were performed with ImageJ (Fiji, version 2.14) using the Cell Counter. The number of SOX2, BRA and TBX6 positive cells was normalized to DAPI. The number of glia cells was quantified as a measure of the total GFAP positive area at day 50 and day 100. For this purpose, a threshold was used to define the total fluorescence area in each image. The same criteria were used to quantify the number of cells expressing LHX3, FOXP1 HOXC6, ISL1, SOX1 and MYF5 and normalized to the total area of DAPI.

To quantify the number of NMJs, α-bungarotoxin staining was performed to label AChR clusters. The cells were fixed with 4% PFA and incubated with Alexa Fluor 647 conjugated α-bungarotoxin (Thermo Fisher) for 2 hours. Images were acquired with 40-80x zoom in 3 random locations per sample ($n = 3$–5). To assess cluster number, AChR clusters (>2 µm) close to neurites were quantified for each image and then normalized to the number of Myosin Skeletal Fast positive fibers present in the image.

Data are reported as the mean ± standard deviation, using a significance level of $p < 0.05$. The number of replicates is indicated in the figure legends. "N" denotes the number of independent experiments and "n" denotes the number of technical replicates/areas analyzed for each sample. With different shapes are denoted the different experiments within the same PSC line. A minimum of three random areas were analyzed in each sample. Data were analyzed by one-way and two-way ANOVA, using Bonferroni test for multiple comparisons and Welch's test for pairwise comparisons using GraphPad Prism 9. Significant differences are indicated as *$p \le 0.05$; **$p \le 0.01$; ***$p \le 0.001$; ****$p \le 0.0001$.

## Reporting summary

Further information on research design is available in the Nature Portfolio Reporting Summary linked to this article.

# Data availability

Further information and requests for resources and reagents should be directed to and will be fulfilled by the Lead Contact, Mina Gouti (mina.gouti@mdc-berlin.de). Source data are provided within this paper..

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

## Acknowledgements

We would like to thank for the initial experiments with NMPs, Marie Sophie Bittner and for excellent technical support Nicole Grieger, Noelle Findeisen and Michelle Müller. We thank Heiko Lickert and Ralf Kühn, who kindly provided us with the XM001 iPSC line. For critical reading of the manuscript, we would like to thank Anthony Gavalas. The work was supported by the Max Delbrück Center (MDC), which receives its core funding from the Helmholtz-Association. M.G. is funded by the European Research Council (ERC) under the European Union's Horizon 2020 research and innovation program (101002689), the Einstein Stiftung Berlin (Einstein Center 3 R, EZ-2020-597-2), and the European Molecular Biology Organization Young Investigator program award. L.V.N.N. was supported by an ECRT PhD fellowship. The generation of SMA iPSCs was supported by NIH R01NS091749 to W.R. Electrophysiology and optogenetic experiments were supported by B.R.R. and D.S. B.R.R. is supported by the Deutsche Forschungsgesellschaft (DFG; project 327654276 - SFB1315 and project 273915538 - SPP1926). D.S. is supported by the DFG (project 184695641 – SFB 958, project 327654276 – SFB 1315, project 415914819 – FOR 3004, project 431572356, the German Excellence Strategy EXC-2049-390688087, NeuroCure), by the European Research Council (ERC) under the European Union's Horizon 2020 research and innovation program (BrainPlay Grant agreement No. 810580) and by the Federal Ministry of Education and Research (BMBF, SmartAge – project 01GQ1420B). For scientific illustrations, we would like to thank SciArtWork.

## Author contributions

M.G. conceived the project, the experimental design, supervised the study and wrote the manuscript. A.U. and M.G. established the 2D soNMJ model. A.U. designed, performed the experiments, data analysis and interpretation, prepared the figures and revised the manuscript. I.L. performed experiments, data analysis for the revisions, revised figures and revised the manuscript. L.V.N.N. performed experiments and data analysis for the revisions and revised the manuscript. B.R.R. performed and analyzed the patch clamp, optogenetic experiments and revised the manuscript. A.G.P. performed initial experiments with SMA lines and revised the manuscript. N.L. performed molecular biology experiments and analysis. G.J.B., H.C.P., M.E.M.G, and W.R. provided the SMA patient 1 iPSC line. W.R. revised the manuscript. S.D. performed the banking, characterization, and quality checks of the SMA patient lines. S.K. performed the TEM experiments. D.S. provided reagents, equipment and support for patch clamp and optogenetic experiments.

## Funding

## Competing interests

MG and AU are co-inventors in patent applications related to the generation of a self-organizing neuromuscular junction cell culture model

filed in USA (18/335,765); Singapore (10202301703 V); Canada (P4410CA). The remaining authors declare no competing interests.
