## [Peer Review File · Nature Communications]

Efficient generation of a self-organizing neuromuscular junction model from human pluripotent stem cellsREVIEWER COMMENTS

Reviewer #1 (Remarks to the Author):

This manuscript describes a bottom-up approach by selecting the appropriate reagents to generate self-organizing 2D cultures of skeletal muscle fibers and brachial motor neurons from human iPSC lines. The developed method results in cultures that can be maintained in culture for up to 100 days. The authors show that stimulation of the soNMJ cultures results in the synchronous contraction of the muscle fibers. Finally, they also use induced pluripotent stem cells from SMA1 patients in the system and show that the soNMJ cultures differ from the controls.

Overall, I find this, in principle, an interesting paper as it provides a unique and quite comprehensive method to create a neuromuscular preparation in a 2D culture. The data on the optogenetic stimulation of motor neurons and the subsequent contraction of the muscle fibers is particularly interesting. If the system can really create functional NMJs, this would be a huge step forward. As the paper stands now, this point is not shown in all aspects. For example, do these synapses really resemble those in vivo? Electron microscopy would help to address this. Are the postsynapses aligned with the nerve terminal; do they contain all of the known components. Moreover, a more in-depth electrophysiological characterization would certainly be helpful.

In addition, the presentation, the quite imprecise wording and the lack of quantification make it often difficult to fully grasp the significance of the finding. Moreover, the authors tend to overstate the impact of the paper. In particular, there are simply no data in the paper that justify the last two sentences of the abstract "This soNMJ model successfully recapitulates the early pathology of spinal muscular atrophy, opening up the opportunity for high throughput studies in disease modeling and drug development. Thus, this model will address unmet needs in the neuromuscular disease field." The SMA data simply suggest that SMA skeletal muscle is impaired; a notion that is already well established. There are no data showing that the NMJ formation is abrogated (involving the motor neurons). All the data can be explained by a muscle-intrinsic problem. In addition, there are no data showing that this model can be used in high throughput studies. It is also unclear whether the soNMJ differentiation protocol can be used consistently and reproducibly, as the authors have already had to adapt the protocol for the cells from SMA patient 2. This use of different protocols makes the subsequent comparisons of the soNMJs difficult.

I also find the discussion not very visionary. I would have hoped that the authors put this into a perspective with other systems that are already published.

Issues:

a. Overall clarity:

The entire paper needs a careful, re-reading and re-editing as it is often quite imprecise and hard to follow. Also the figure legends need complete re-writing as the legends do not allow to understand the experiments. Some of the symbols are also not introduced (e.g., Fig. 3d, Fig. 5c).

b. Quantification:

RT-qPCR of Fig. 1b is a mix of different iPSC lines. While it is stated in legend that there were (XM001: N=3, n=9; H1: N=1, n= 3; H9: N=1, n=3) samples, some of the graphs do not include data from H9 (all the NP genes); MYF5 data include only N=2 for XM001 but N=2 for H1. Similar issues are seen in Fig. 2e. If this is not based on sloppiness, the authors need to explain why they selected some of the groups.

On line 152, the authors write “whereas the expression of the early MN marker ISL1/2 was completely abolished (Fig. 2f).” There are no data on ISL1/2 in Fig. 2f.

In Fig. 2e, the relative proportion of medial and lateral motor column neurons are calculated. The total proportion of brachial MN is around 80% of which 73% are medial and 25% are lateral. These numbers do not add up. Any explanation for this?

c. Writing:

Line 199: You mention that the PAX7+ cells are underneath the basal lamina. This is not seen in Suppl. Fig. 5e. Moreover, the term basal lamina refers to a structured extracellular matrix (EM term). I am sure that ECM in culture does not form a structure that deserves to be called basal lamina.

Line 217 ff: Fig. 3e does not show an acceleration of contraction but an increase in amplitude. In fact, contraction is rather decelerated.

d. Detailed comments:

Line 4ff: Sentence does not make sense; some repeat of “spinal cord neurons, skeletal muscles, glia and terminal Schwann cells”

line 33: “these neuromuscular cultures started contracting at day 50” – only the fibers contract – There is no early time point, so this is an overstatement – spontaneous contractions occur in cultures without neurons

line 163: “in vivo” should be italic

line 189/190: “we detected numerous AChR clusters by staining for alpha-bungarotoxin contacted by TUBB3+ neurites, suggesting the formation of neurites” are all AChRs co-localizing with neurites? Or is the distant secretion of agrin by neurons sufficient to induce AChR clustering without contact with neurites?

line 229 ff: “Moreover, repetitive injections of short (10 ms) depolarizing currents at 10 Hz reliably elicited trains of action potentials (Fig. 4c).” should be Fig. 4b.

line 242: The calcium waves need to be indicated in the video to make the point; Figure 4f is not much telling.

lines 266/Figure S6a: No quantification of SMN protein levels except by staining; Western blot analysis would be preferred.

line 267/268: “ We further used the SMA lines to generate soNMJs in vitro through an NMP intermediate state (Supplementary Fig. 5b,c).” : “in vitro” in italic and these are actually Supplementary Figures 6b and 6c

line 270: “DESMIN+ myoblast cells and SOX1+ NPs were detected”. However, this is not shown. Please show.

Supp Fig 6c: something seems wrong here: is there no control; what is the meaning of the different colors? Where are the hiPSCs datapoints for TBXT qPCR (SMA pt 2)

line 287: "higher abundance of MNs in SMA... might be due to a longer proliferation phase resulting from differentiation failure". There are no data to support this hypothesis; also no reference is cited that indicates that this may happen

line 297 ff: "Acetylcholine administration did not stimulate contraction, whereas curare failed to stop spontaneous muscle activity, suggesting the lack of functional NMJs in the SMA model (Fig. 5f, Video 5)." This could also be based on a problem of skeletal muscle fibers per se without the involvement of motor neurons as muscle fibers are also SMN negative.

e. Figures:

Besides the issues mentioned above, I also did not at all like the drawing of the motor neuron innervating the muscle as it is scientifically wrong. Each fiber is innervated by one motor neuron and NMJs are not at the tendon site but often aligned in the middle of a muscle. In some cases, the nerve terminals are not even touching the muscle (Fig. 1a, d; Fig. 3a; Fig. 4d; Fig. 5).

Fig. 5f: What is the difference between contraction and contraction amplitude?

f: General minor remark:

The numbering of the videos is odd. In the text, they are numbered; in the material provided, they have some random names.

Reviewer #2 (Remarks to the Author):

In this paper, Urzi et al. reported a protocol for inducing two-dimensional self-organized neuromuscular junctions (soNMJs) from induced pluripotent stem cells (iPSCs) via neural mesoderm progenitor cells (NMPs). They further showed that many motoneurons (MNs) in this differentiation method expressed markers of the median motor column of the brachial spinal cord, suggesting that the soNMJ method allows for the induction of position-specific NMJs. Overall, the paper is written in precise English, and there are no logical lapses. The quality of the immunostaining figures and electrophysiological analysis is also excellent.

However, a method to construct 2D NMJs from iPS cells in a short period of time in an autologous manner has already been reported. In addition, that report has already modeled human NMJ lesions in spinal muscular atrophy (Lin CY, JCI insight 2021). Therefore, the advantage of this study is that the authors accurately induced NMPs and confirmed their position specificity. However, it is also known that NMPs can induce both neurons and skeletal muscle cells. Therefore, the scientific significance of the paper is limited.

Major comments

Although it is clear from electrophysiological experiments that soNMJ is functional, morphological evaluation of NMJ is lacking. This is important for estimating the maturity of NMJs. In mature human NMJs, acetylcholine receptor (AChR) clustering in the postsynaptic membrane aligns with MN axon terminal, forming the nummular shape (coin-shaped patches) (Jones RA. Cell Reports 2017). Higher resolution images than Figure 3C should be obtained to confirm typical NMJ morphologic features.

The postsynaptic accumulation of synaptic vesicle and mitochondria, and active zone in axon terminal, which are typical structure of matured NMJ, also should be observed with transmission electron microscopy.

In Supplementary Figure 3b, there appears to be a significant difference in differentiation propensity among H1 and the other clones. This is especially significant for the expression of OLIG1, ILSL1, ChAT, and MYOD, raising a question about the robustness of the system. Even in the differentiation propensity toward PSM in Figure 1b, H1 clearly differs from the other clones. In addition, there are many experiments with little biological replication (N=1). Therefore, the reviewer suggests to increase N or clones to verify the robustness of the system.

The authors state that one of the main aims of constructing 2D-soNMJ is to apply iPS cell-derived NMJs for high-throughput studies, and claim that current differentiation system is useful for high-throughput studies. However, despite the very small number of experimental replications in some data, there is a large variation in differentiation efficiency (such as in Figure 2b), which raises question whether HTS is possible with this method. If the authors wish to demonstrate the potential usefulness of soNMJ in HTS, it is necessary to show the robustness and stableness of their soNMJ differentiation in a HTS platform such as on 96 well-plate culture.

Terminal Schwann cells should cover most of the NMJ (Alhindi, A, Brain Communications 2021). In Supplementary Figure 5F, to demonstrate the interaction between Schwann cells and NMJ, author should show that Schwann cells indeed cover the NMJ by TEM or high-magnification immunostaining.

In Figure 3d and Figure 5c, the authors evaluated the number of NMJs with aBTX/MyHC. However, as mentioned above, the shape of the aBTX clusters is very poor, so it is possible that they overestimate the number of NMJs by including spontaneous clustering on muscle fibers. The authors should evaluate the number and areas of aBTXs merging with motoneuron terminals.

Minor comments

For immunostaining, please describe in Figure Legends which clones were used in each figure.

The authors stated that skeletal muscle and neurons are spatially co-localized in layers along the z-axis in Video S1, but the two are clearly spatially separated in Figure 3b. Please explain why the discrepancy between the two occurs.

In Figures 2c and 2e, did the authors also check the negative expression of HOX genes expressed in other somites?

For the SMA model, the authors showed an obvious skeletal muscle dysplasia (Figures 5b-5d). This result is consistent with numerous reports of cellular autonomous skeletal muscle abnormalities in SMA. However, given this substantial skeletal muscle dysfunction, it is likely that the reduced muscle contractility in Figure 5f is primarily due to skeletal muscle dysfunction, which makes it difficult to assessment of the function of the SMA-NMJ.

In Supplementary Figure 5e, it is difficult to confirm that satellite cells localize underneath laminin. To show that, please provide 3D reconstructed image from a scan of the Z-axis.

Reviewer #3 (Remarks to the Author):

The manuscript of Urzi et al. describes the generation of a new neuromuscular junction model based on the differentiation of human pluripotent stem cells into neuromesodermal progenitors and then the capacity of these progenitors to self-organize into neuromuscular systems. The possibility to maintain these cultures up to 100 days leads to the generation of models that appear well matured and functional. The authors also demonstrate the application of this system to a neuromuscular disease, the Spinal Muscular Atrophy, that is characterized by the degeneration of spinal motoneurons. Despite a growing number of studies describing the generation of a NMJ model from human pluripotent stem cells, the possibility to generate human NMJ model in 2D maintained for 100 days is really interesting and novel. Although the data are presented effectively, they are major concerns that required revision and mentioned below.

1. The description of the protocol in Figure 1A as well as in the text (line 85) is not totally clear and should be improved. As indicated by the authors (line 85), « to enhance the generation of both cell types, we continued the treatment with WNT/ bFGF/2SMADi which does not correspond to the Figure 1A.

2. In Supplementary Figure 2b-c, could the authors confirm that all the cells TBXT positive are also SOX2 ? what is consequently the proportion of SOX2 positive cells that do not express TBXT ?

3. In Figure 1c, the authors suggest segregation between NP and PSM. First, what is the reproducibility of these results among the different cell lines? Could the authors also provide quantification of the proportion of PAX3 and SOX2 positive cells?

4. Regarding this segregation observed at Day 6, which is a central point of this manuscript, could the authors explain why they decided to passage the cells at the same time point (day 6) as indicated line 106 ? It is indeed difficult to understand how this segregation can be maintained after dissociation. Could the authors comment?

5. Could the authors also explain the choice of the concentrations used for RA and SAG? Did they optimize these concentrations as it has been previously demonstrated that the concentration of RA

might influence the proportion of hPSC-derived spinal MNs (Maury et al., 2015).

6. In Figure 1d, the protocol as well as the characterization of hiPSC-derived spinal MNs are not clearly explained. What is the protocol used to generate spinal MNs? Immunostaining for TUBB3 is not adapted to evaluate the number of spinal MNs and ISL1/ILS2 or Hb9 should be considered. The same concern can be raised for the characterization of MNs obtained in the cocultured system.

7. Line 114, could the author explain the rationale to maintain the cultures only in the presence of HGF and IGF ?

8. Regarding the characterization of the NMJ models at day 20, the authors quantified the expression of Olig2 only by RT-qPCR. Could they indicate the proportion of OLIG2 positive cells in order to determine the efficiency of DAPT treatment ? This comment is also important as from Supplementary Figure 3b, the expression of OLIG2 appears really variable among the different cell lines. Could the authors comment?

9. In the same line, could the author comment why ISL1 mRNA is observed only in H1 line (Supplementary Figure 3b) whereas ISL1 positive cells are also observed in XM001 line at the same time ? Did the authors look at other markers of MNs such as Hb9 ?

10. To determine the identity of spinal MNs, the use of HOXC6 only is not sufficient. To clearly determine the brachial identity, the authors should analyze the presence of other HOX genes, such as HOXA5 and HOXC8. Other HOX genes such as HOXD9 or HOXC10 should be analyzed in order to exclude the presence of other subtypes.

The proportion of HOXC6 positive cells among the ISL1 population seems to be also very variable from 40 to 100%. This variability is also observed for the ISL1 positive cells (18,9% \pm 10.1 as indicated line 125). It is therefore difficult to understand the statement of an « efficient » generation. Could the authors comment and explain why they have such variability even with the same cell line ? It could be interesting to also indicate the proportion of neurons (by using a pan neuronal marker such TUBB3) in order to estimate the proportion of spinal MNs within the neuronal population?

11. As the above comment, a better quantification of the proportion of MNs, glial cells and interneurons generated should be provided at later stage of the protocol (Day 50 and D100) ? Only examples of immunostaining are provided for the moment except for GFAP for which a quantification of the staining area is provided. Thus, a more precise quantification of the positive cells should be provided in Supplementary Figure 4. This should help at evaluating the reproducibility of the protocol. In addition, regarding the quantification of staining area, it is difficult to accurately evaluate the results as the normalization of this quantification is not indicated. The authors should indicate on which parameters they normalized this quantification ? per field ? per number of DAPI positive cells ?

12. Due to the variability described in Figure 2, it appears necessary to indicate if such variability is also observed at day 50 and day 100. Especially, can the beautiful and impressive image in Figure 3b be observed with the other cell lines ?

13. In Supplementary Figure 5d, how did the authors normalize the quantification of the NMJ length ?

What is the proportion of the index fusion (meaning the number of myotubes with more than 3 nuclei) ? Did the authors normalize the AChR clustering with the size of myotubes ?

14. In figure 3D a similar concern can be raised. What does BTX/ Fast MyHC represent? the number of BTX aggregates per Fast MyHC ? did the authors select the size of BTX aggregates ? Did the authors select the Fast MyHC containing more than 3 nuclei ?

15. Regarding the functionality, the authors demonstrated that the contractions can be blocked by curare treatment that act at the post-synaptic level. In order to clearly demonstrate the connectivity, the authors should demonstrate the blockage of the contractions using Botulinium Toxins as these drugs will act directly on the synaptic transmission from the MNs.

16. The authors claim « in those cultures, the contraction of the skeletal muscles could not be maintained for too long due to loss of attachment. An important advantage of the soNMJ model is that muscles remained attached to the dish for long periods, even under contractile conditions (Supplementary Fig. 5a,c). This is an interesting observation but could the authors comment or discuss this result? Did they observe more focal adhesions that increase muscle surface interactions ?

17. At day 50 and 100, are the NMJs as mature as in neuromuscular organoids? In Faustino M et al, the group presented TEM images of synaptic clefts, do the authors have such organization in their soNMJ models?

18. The optogenetic experiments are very elegant. However, what is the proportion of motoneurons transduced by the different AAV used for these experiments (Figure 4)?

19. It could be also elegant to use optogenetic stimulation in the context of SMA. Indeed, the authors claim an early NMJ defects in SMA condition in comparison with control. The use of optogenetic will reinforce these conclusions which for the moment rely only on the quantification of NMJ.

20. Line 284, the authors concluded that « Unexpectedly, the abundance of MNs appeared higher in SMA than in control conditions ». These results are very surprising and should be better characterized. Thus, as already mentioned for the control condition, a better quantification of the number of MNs should be determined.

21. Similarly, a decrease in NMJ is observed but as also described by the authors SMA myotubes are smaller (line 278). This observation might explain the decrease in NMJ. Again, it should be informative to know how many skeletal muscle cells are present both in SMA and control conditions. In addition, an isogenic control should be included as strongly recommended by the ISSCR guidelines. Thus, the re-introduction of SMN gene should be performed in SMA line in order to analyze

Reviewer Response Letter

We would like to thank the reviewers for their valuable feedback on our manuscript. We have carefully considered all the comments and made the necessary revisions to address each point raised. Below is a detailed response to each comment along with the corresponding changes made in the revised manuscript.

Reviewer #1 (Remarks to the Author):

This manuscript describes a bottom-up approach by selecting the appropriate reagents to generate self-organizing 2D cultures of skeletal muscle fibers and brachial motor neurons from human iPSC lines. The developed method results in cultures that can be maintained in culture for up to 100 days. The authors show that stimulation of the soNMJ cultures results in the synchronous contraction of the muscle fibers. Finally, they also use induced pluripotent stem cells from SMA patients in the system and show that the soNMJ cultures differ from the controls.

Overall, I find this, in principle, an interesting paper as it provides a unique and quite comprehensive method to create a neuromuscular preparation in a 2D culture. The data on the optogenetic stimulation of motor neurons and the subsequent contraction of the muscle fibers is particularly interesting. If the system can really create functional NMJs, this would be a huge step forward. As the paper stands now, this point is not shown in all aspects. For example, do these synapses really resemble those in vivo? Electron microscopy would help to address this. Are the postsynapses aligned with the nerve terminal; do they contain all of the known components. Moreover, a more in-depth electrophysiological characterization would certainly be helpful.

We thank the reviewer for the supportive comments. We have performed additional experiments and included a new supplementary figure where we analyzed the soNMJ model by electron microscopy (Supplementary Fig. 6) to show the mature organization of skeletal muscle, the folding of the membrane, the alignment with neural terminals, the presence of mitochondria, t-tubules, and synaptic vesicles. We have also performed additional optogenetic experiments to characterize the SMA soNMJ model (Supplementary Fig. 8f, Video 8).

In addition, the presentation, the quite imprecise wording and the lack of quantification make it often difficult to fully grasp the significance of the finding. Moreover, the authors tend to overstate the impact of the paper. In particular, there are simply no data in the paper that justify the last two sentences of the abstract "This soNMJ model successfully recapitulates the early pathology of spinal muscular atrophy, opening up the opportunity for high throughput studies in disease modeling and drug development. Thus, this model will address unmet needs in the neuromuscular disease field." The SMA data simply suggest that SMA skeletal muscle is impaired; a notion that is already well established. There are no data showing that the NMJ formation is abrogated (involving the motor neurons). All the data can be explained by a muscle-intrinsic problem. In addition, there are no data showing that this model can be used in high throughput studies. It is also unclear whether the soNMJ differentiation protocol can be used consistently and reproducibly, as the authors have already had to adapt the protocol for the cells from SMA patient 2. This use of different protocols makes the subsequent comparisons of the soNMJs difficult.

We have rewritten the last part of the abstract and performed additional experiments including extensive quantifications of the various cell types at different developmental stages. To show how the soNMJ model can be used for high throughput experiments, we have also performed analysis of the SMA soNMJ model in 96-well plate format and quantified the difference in the neural and muscle areas (Supplementary Fig. 8). Regarding the SMA modelling, we agree with the reviewer that part of the phenotype can be explained by a muscle-specific effect but it is important to point out that causality has not been established. To address this, the precise temporal sequence of events needs to be established in a human specific model. The advantage of using such a model is that it allows for greater spatial and temporal resolution of the events that happen *prior* to the disease state. Our data suggest an early muscle phenotype in a human-specific model. This is impossible to establish by analysing post-mortem patient biopsies. In the future, more detailed analysis will be needed to understand the mechanisms by which the muscle cells are affected at the early stages of the disease. While this falls out of the scope of the specific study, we are excited to follow up on this question.

I also find the discussion not very visionary. I would have hoped that the authors put this into a perspective with other systems that are already published.

We have revised the discussion to include the advantages of the specific soNMJ model compared to previously published NMJ models and the important applications that we expect this model will find. Still, we believe that different models can be used to address different questions, and might not be that one model fits all purposes. For example, our previously published NMO model is not ideal for high throughput studies due to the large size of organoids. Thus, for high throughput studies, the soNMJ model will be a better solution where one can screen a large number of compounds and then perform a targeted screen on NMOs. We have included this point in the introduction.

Issues:

a. Overall clarity:

The entire paper needs a careful, re-reading and re-editing as it is often quite imprecise and hard to follow. Also the figure legends need complete re-writing as the legends do not allow to understand the experiments. Some of the symbols are also not introduced (e.g., Fig. 3d, Fig. 5c).

We have edited the manuscript to improve clarity throughout. Additionally, we have rewritten the figure legends to enhance the understanding of the depicted experiments. We have introduced specific colors for each hPSC line and different shapes to differentiate between experiments in all figures. All changes are marked in the manuscript with blue color.

b. Quantification:

RT-qPCR of Fig. 1b is a mix of different iPSC lines. While it is stated in legend that there were (XM001: N=3, n=9; H1: N=1, n=3; H9: N=1, n=3) samples, some of the graphs do not include data from H9 (all the NP genes); MYF5 data include only N=2 for XM001 but N=2 for H1. Similar issues are seen in Fig. 2e. If this is not based on sloppiness, the authors need to explain why they selected some of the groups.

We are sorry for the inaccuracy. We have modified Figure 1b and have included qPCR data for all genes at day 6 for Np and Np+PSM conditions from XM001: N=3, n=9; H1: N=2, n=6; H9: N=1, n=3. We have also included the information and corrected the sample size in the figure legend accordingly.

On line 152, the authors write “whereas the expression of the early MN marker ISL1/2 was completely abolished (Fig. 2f).” There are no data on ISL1/2 in Fig. 2f.

The expression of ISL1/2 is downregulated in motor neurons as they mature. At day 50, the expression of Islet1 is downregulated and the motor neurons express ChAT. We think this is not an important point for the paper and removed the relevant sentence. We include here an immunofluorescence image of ISL1 at day 50 soNMJ culture for the reviewer.

Figure R1: Immunofluorescence analysis at day 50 soNMJ model for the motor neuron marker ISL1 in the neural area expressing TUBB3 shows that ISL1 is downregulated.

In Fig. 2e, the relative proportion of medial and lateral motor column neurons are calculated. The total proportion of brachial MN is around 80% of which 73% are medial and 25% are lateral. These numbers do not add up. Any explanation for this?

This was not clear in the manuscript. We have quantified the number of ISL1⁺/ HOXC6⁺ cells and found that they are around 80% of all ISL⁺ MNs. We then quantified the number of MNs that have an LMC or MMC identity. Of all the MNs (100%), 75% have a medial motor column (MMC) identity and 25% a lateral motor column (LMC) identity. We have modified the text to make it clearer. “Analysis with specific columnar identity markers revealed that **within the MN population**, approximately....”

c.Writing:

Line 199: You mention that the PAX7⁺ cells are underneath the basal lamina. This is not seen in Suppl. Fig. 5e. Moreover, the term basal lamina refers to a structured extracellular matrix (EM term). I am sure that ECM in culture does not form a structure that deserves to be called basal lamina.

We agree with the reviewer that we do not have definitive evidence that a basal lamina similar to the *in vivo* forms *in vitro*. We have adjusted the text accordingly and included a 3D reconstruction video of a confocal PAX7/LAMININ imaging to better show the localization of PAX7⁺ cells (Video 6).

Line 217 ff: Fig. 3e does not show an acceleration of contraction but an increase in amplitude. In fact, contraction is rather decelerated.

Regarding the description of Fig. 3e, we have corrected the text to accurately reflect an increase in contraction amplitude rather than acceleration. We apologize for any confusion caused by the previous description and thank the reviewer for pointing it out. The revised text now clearly states that treatment with acetylcholine increased the amplitude of muscle contraction.

d. Detailed comments:

Line 4ff: Sentence does not make sense; some repeat of “spinal cord neurons, skeletal muscles, glia and terminal Schwann cells”

We have modified the text accordingly to remove the repetition that was introduced by mistake.

line 33: “these neuromuscular cultures started contracting at day 50” – only the fibers contract – There is no early time point, so this is an overstatement – spontaneous contractions occur in cultures without neurons.

We appreciate the reviewer's comment and understand the concern regarding the statement about neuromuscular cultures contracting at day 50. To address this concern, we have included additional data in the revised manuscript to demonstrate that the muscle fibers exhibit spontaneous contractions at day 50. We have also performed experiments using curare to block these contractions, providing evidence that the contractions are mediated by the functional NMJs formed in the cultures (Video 2a, 2b, 2c). We have added movies at both day 50 and day 75 to showcase the contraction dynamics and provide a clearer understanding of the timing and progression of the muscle contractions in relation to NMJ formation.

These additional data and experiments provide support for our statement that the neuromuscular cultures start contracting at day 50, which aligns with the detection of functional NMJs. We hope that these data address the reviewer's concern and provide a more accurate representation of the findings.

line 163: “in vivo” should be italic

Corrected

line 189/190: “we detected numerous AChR clusters by staining for alpha-bungarotoxin contacted by TUBB3⁺ neurites, suggesting the formation of neurites” are all AChRs co-localizing with neurites? Or is the distant secretion of agrin by neurons sufficient to induce AChR clustering without contact with neurites?

We appreciate the reviewer's question regarding the AChR clusters and their association with neurites. In our study, we quantified the number of AChR clusters that are in close proximity to TUBB3⁺ neurites. This indicates that the AChR clusters are indeed in contact with neurites, suggesting the formation of neuromuscular junctions (NMJs). This observation supports the notion that the AChR clustering is induced by the contact and signaling between motor neurons and muscle fibers.

Furthermore, to provide additional evidence of the functional connectivity between motor neurons and muscle fibers in our soNMJ cultures, we performed chemical and optogenetic experiments (Fig. 3e, Fig. 4d-f). The results of these experiments demonstrate that optogenetic activation of motor neurons leads to synchronous contraction of the muscle fibers. This further supports the presence of functional NMJs and the establishment of communication between motor neurons and muscle fibers.

Taken together, the co-localization of AChR clusters with neurites and the observed synchronous muscle contraction upon motor neuron activation provide strong evidence for the formation of functional NMJs in our soNMJ cultures. These findings support the notion that our culture system successfully recapitulates key aspects of neuromuscular junction development and function. The question of agrin function in the soNMJ system is important and will be the focus of more detailed mechanistic studies in the future but we think that it falls outside the scope of the present manuscript.

line 229 ff: “Moreover, repetitive injections of short (10 ms) depolarizing currents at 10 Hz reliably elicited trains of action potentials (Fig. 4c).” should be Fig. 4b.

We have checked and corrected all the figure panel numbers in the revised version of the manuscript.

line 242: The calcium waves need to be indicated in the video to make the point; Figure 4f is not much telling.

We have included arrows with two different colors green (neurons) and purple (muscles) that highlight the calcium waves in videos 3a and 3b.

lines 266/Figure S6a: No quantification of SMN protein levels except by staining; Western blot analysis would be preferred.

We appreciate the reviewer's suggestion to include quantification of SMN protein levels using Western blot analysis. In response to this suggestion, we have performed Western blot analysis to measure SMN protein levels in both control and SMA patient lines. The results of the Western blot analysis are now included in Supplementary Figure 7b, providing quantitative data on the lower levels of SMN protein in the SMA patient lines compared to the controls.

line 267/268: “ We further used the SMA lines to generate soNMJs in vitro through an NMP intermediate state (Supplementary Fig. 5b,c).” : “in vitro” in italic and these are actually Supplementary Figures 6b and 6c

Corrected as suggested.

line 270: “DESMIN+ myoblast cells and SOX1+ NPs were detected”. However, this is not shown. Please show.

The immunofluorescence analysis of DESMIN and SOX1 at day 20 can be found in Supplementary Fig. 7e.

Supp Fig 6c: something seems wrong here: is there no control; what is the meaning of the different colors? Where are the hiPSCs datapoints for TBXT qPCR (SMA pt 2)

We thank the reviewer for spotting the error. This was an earlier version and we have now updated the plot and included all the datapoints in Supplementary Fig. 7c. The meaning of the different colors is now explained in the figure legend.

line 287: “higher abundance of MNs in SMA... might be due to a longer proliferation phase resulting from differentiation failure”. There are no data to support this hypothesis; also no reference is cited that indicates that this may happen

We appreciate the reviewer's comment regarding the speculative statement about the longer proliferation phase of SMA motor neurons. We have removed this sentence. We performed high throughput analysis of the control and SMA line in 96 well plates and analyzed the difference in the number of neurons, MNs, and skeletal muscle cells. Apparently, there is no statistically significant difference in neurons and MNs whereas the number of muscle cells is significantly reduced. The data are included in a new Supplementary Fig. 8.

line 297 ff: “Acetylcholine administration did not stimulate contraction, whereas curare failed to stop spontaneous muscle activity, suggesting the lack of functional NMJs in the SMA model

(Fig. 5f, Video 5).” This could also be based on a problem of skeletal muscle fibers per se without the involvement of motor neurons as muscle fibers are also SMN negative.

We agree with the reviewer's comment and acknowledge that the lack of functional NMJs in the SMA model could potentially be attributed to a problem in the skeletal muscle fibers themselves, as they are also SMN-negative. In response to this concern, we have conducted additional experiments using optogenetic activation of motor neurons in the SMA patient NMJ model (Supplementary Fig. 8f, Video 8). These experiments demonstrate that despite the presence of motor neurons, the synchronous contraction of skeletal muscle is not observed upon optogenetic stimulation, indicating a connectivity problem between the motor neurons and skeletal muscles. These findings support the notion that the observed phenotype in the muscle and NMJs is not solely due to muscle fiber impairment but involves a defect in the connectivity between motor neurons and skeletal muscles. We have included these optogenetic experiments in the revised manuscript to provide further evidence (Supplementary Fig. 8f, Video 8). In the Video, it is apparent that while the muscle has some spontaneous activity, there is no synchronous contraction due to the optogenetic stimulation.

e. Figures:

Besides the issues mentioned above, I also did not at all like the drawing of the motor neuron innervating the muscle as it is scientifically wrong. Each fiber is innervated by one motor neuron and NMJs are not at the tendon site but often aligned in the middle of a muscle. In some cases, the nerve terminals are not even touching the muscle (Fig. 1a, d; Fig. 3a; Fig. 4d; Fig. 5).

We appreciate the reviewer's feedback regarding the illustration of the motor neuron innervating the muscle in our manuscript. We acknowledge that the previous illustration was oversimplified and not scientifically accurate in depicting the innervation pattern of motor neurons and the localization of NMJs. We have taken the reviewer's comment into consideration and have modified the image to better represent the actual connectivity between motor neurons and muscles. We hope that the revised image provides a more accurate representation.

Fig. 5f: What is the difference between contraction and contraction amplitude?

In the context of Fig. 5f, the term "contraction" refers to the occurrence of a muscle contraction, while "contraction amplitude" is the actual measure of each contraction spike. We used contraction amplitude in the plot to remove the noise that was introduced by the movement in the contraction graph. By analyzing and plotting these contraction amplitudes over time, it becomes possible to observe the pattern of muscle activity while minimizing the impact of noise introduced by movement artifacts (out-of-focus images).

f: General minor remark:

The numbering of the videos is odd. In the text, they are numbered; in the material provided, they have some random names.

We are sorry about the random names. We have changed all the names in the movies and aligned them with the names in the text.

Reviewer #2 (Remarks to the Author):

In this paper, Urzi et al. reported a protocol for inducing two-dimensional self-organized neuromuscular junctions (soNMJs) from induced pluripotent stem cells (iPSCs) via neural mesoderm progenitor cells (NMPs). They further showed that many motoneurons (MNs) in this differentiation method expressed markers of the median motor column of the brachial spinal cord, suggesting that the soNMJ method allows for the induction of position-specific NMJs. Overall, the paper is written in precise English, and there are no logical lapses. The quality of the immunostaining figures and electrophysiological analysis is also excellent.

However, a method to construct 2D NMJs from iPS cells in a short period of time in an autologous manner has already been reported. In addition, that report has already modeled human NMJ lesions in spinal muscular atrophy (Lin CY, JCI insight 2021). Therefore, the advantage of this study is that the authors accurately induced NMPs and confirmed their position specificity. However, it is also known that NMPs can induce both neurons and skeletal muscle cells. Therefore, the scientific significance of the paper is limited.

We appreciate the reviewer's positive feedback on the clarity of the manuscript and the quality of the experimental analysis. We also acknowledge the reviewer's comment regarding the existence of a previously reported method for constructing 2D NMJs from iPSCs and modeling human NMJ lesions in spinal muscular atrophy. While it is true that a method for constructing 2D NMJs from iPSCs has been reported, the authors in the specific study use the inducible expression of MYOD to generate the muscle cells and NMJs become functional only after optogenetic stimulation (Lin, Yoshida et al. 2019). While this is an interesting model, we would like to emphasize the unique contributions of our study. Our study focuses on the accurate induction of neuromesodermal progenitor cells (NMPs) and the confirmation of their position-specific differentiation into motoneurons expressing markers of the median motor column and lateral motor column of the brachial spinal cord. In our study, spinal cord neurons, skeletal muscle cells, glia, and terminal Schwann cells are generated concomitantly in the absence of transgenes resembling the in vivo process. Furthermore, the NMJs form and are functional even in the absence of optogenetic stimulation. Thus, there are unique advantages in the soNMJ model compared to other models.

Regarding the concern about NMPs being capable of inducing both neurons and skeletal muscle cells, we acknowledge that this is a known feature of NMP differentiation. However, achieving the precise differentiation and maturation of these cells into functional NMJs is a complex process that required the careful optimization of culture conditions and signaling cues. Our study addresses these challenges and provides a detailed protocol for the induction of position-specific NMJs, demonstrating the significance of our work. We believe that the induction of position-specific NMJs and the establishment of a two-dimensional self-organized system provide valuable contributions to the field and offer new opportunities for studying motor neuron, skeletal muscle development, and disease modeling.

Major comments

Although it is clear from electrophysiological experiments that soNMJ is functional, morphological evaluation of NMJ is lacking. This is important for estimating the maturity of NMJs. In mature human NMJs, acetylcholine receptor (AChR) clustering in the postsynaptic membrane aligns with MN axon terminal, forming the nummular shape (coin-shaped patches)

(Jones RA. Cell Reports 2017). Higher resolution images than Figure 3C should be obtained to confirm typical NMJ morphologic features.

We appreciate the reviewer's comment and agree that morphological evaluation is important for assessing the maturity of NMJs. We have now included a 3D image of the NMJ in Figure R2 for the reviewer and the 3D video in the manuscript (Video 7).

The postsynaptic accumulation of synaptic vesicle and mitochondria, and active zone in axon terminal, which are typical structure of matured NMJ, also should be observed with transmission electron microscopy.

We appreciate the suggestion made by the reviewer. To address this, we have performed electron microscopy (EM) analysis of the soNMJ at day 75. We have included a new Electron Microscopy Supplementary Fig. 6, which shows the organization of the muscle, folding of the basal membrane, the presence of synaptic vesicles, mitochondria and t-tubules. The EM images provide further evidence of the matured NMJ structures.

In Supplementary Figure 3b, there appears to be a significant difference in differentiation propensity among H1 and the other clones. This is especially significant for the expression of OLIG1, ISL1, ChAT, and MYOD, raising a question about the robustness of the system. Even in the differentiation propensity toward PSM in Figure 1b, H1 clearly differs from the other clones. In addition, there are many experiments with little biological replication (N=1). Therefore, the reviewer suggests to increase N or clones to verify the robustness of the system.

We appreciate the reviewer's concern about the differentiation propensity among different clones and the need for increased biological replication. To address this, we have increased the number of experiments and included all the information in the source data where we summarize the experiments performed with each hPSC line. In the source data, we have also included a separate sheet named "immunofluorescence_cell_lines", where we summarize all the different lines that were used for the immunofluorescence experiments. We agree with the reviewer that there are differences in the response between the different human cell lines, which stems to the different genetic background, a notion well established in the field (Osafune, Caron et al. 2008, Bock, Kiskinis et al. 2011, Ortmann and Vallier 2017). However, we show here that the overall trends and conclusions remain valid based on the replication and similar response of all cell lines (Supplementary Fig. 3b) and the observed consistency in the formation of NMJs among different human PSC lines (Fig. 3d).

The authors state that one of the main aims of constructing 2D-soNMJ is to apply iPS cell-derived NMJs for high-throughput studies, and claim that current differentiation system is useful for high-throughput studies. However, despite the very small number of experimental replications in some data, there is a large variation in differentiation efficiency (such as in Figure 2b), which raises question whether HTS is possible with this method. If the authors wish to demonstrate the potential usefulness of soNMJ in HTS, it is necessary to show the robustness and stableness of their soNMJ differentiation in a HTS platform such as on 96 well-plate culture.

We appreciate the reviewer's comment and agree that demonstrating the robustness and stability of the soNMJ differentiation method is crucial for its potential use in high-throughput studies. To address this concern, we have now performed many of the experiments in 96-well plates and analyzed them using the high-content imaging system Opera Phenix. We have

included this information in the Materials and Methods section and Supplementary Fig. 8 to emphasize the potential for scalability and high-throughput applications of the soNMJ method.

Terminal Schwann cells should cover most of the NMJ (Alhindi, A, Brain Communications 2021). In Supplementary Figure 5F, to demonstrate the interaction between Schwann cells and NMJ, author should show that Schwann cells indeed cover the NMJ by TEM or high-magnification immunostaining.

We appreciate the reviewer's suggestion. To demonstrate the interaction between Schwann cells and NMJs, we have included a 3D image for the reviewer below and a new movie in the manuscript (Video 7) showing the 3D reconstruction of the NMJs with S100 β , a Schwann cell marker, TUBB3, and aBTX. The video better illustrates the coverage of NMJs by Schwann cells and highlights their interaction. We would like to emphasize that there are fundamental differences between the mouse and human NMJ as the reviewer mentioned. These differences are also true for terminal Schwann cells which in the human NMJs tend to have more non-synaptic placement of their nuclei and less coverage of neighboring AchRs (Alhindi et al, *Brain Commun*, 2021).

Figure R2: 3D projection of a confocal soNMJ image stained at day 100 with S100 β ; green/TUBB3; magenta/aBTX; cyan. Long NMJs (cyan) are observed in close proximity with S100b (green) terminal Schwann cells.

In Figure 3d and Figure 5c, the authors evaluated the number of NMJs with aBTX/MyHC. However, as mentioned above, the shape of the aBTX clusters is very poor, so it is possible that they overestimate the number of NMJs by including spontaneous clustering on muscle fibers. The authors should evaluate the number and areas of aBTXs merging with motoneuron terminals.

We have quantified the number of AchR clusters that are close to neurites. We have now included this information in the methods section. It is very challenging to quantify the number of NMJs in the manner suggested due to the thickness of the cultures. The important piece of information here would be the *difference* between wt and affected cultures (e.g. from SMA iPSCs) and not so much the absolute number of NMJs. Furthermore, it is clear that there is functional connectivity between the motor neurons and skeletal muscles based on the drug

experiments (Fig. 3d) and optogenetic experiments presented in Fig. 4d, 4f. Additionally, contraction is dependent upon NMJs, as the curare experiments show, further supporting our conclusion that these NMJs are fully functional.

Minor comments

For immunostaining, please describe in Figure Legends which clones were used in each figure.

We have included in the source data a separate sheet named “immunos_cell_lines” where we describe for all figures/panels the clones that were used for the analysis. We have also included the information in the figure legends.

The authors stated that skeletal muscle and neurons are spatially co-localized in layers along the z-axis in Video S1, but the two are clearly spatially separated in Figure 3b. Please explain why the discrepancy between the two occurs.

In Fig. 3b, we show the spatial segregation of neurons and muscles at day 50 on the xy-axis. In Video S1, we analyzed the cultures at day 100, where a large number of axons are on top of the skeletal muscle cells, and we show that even in that case, the neurons are on a different layer in the z-axis. Overall, the cultures segregate on the xy and z-axis, and this is what we wished to illustrate in the manuscript by including the image in Fig. 3b and Video 1.

In Figures 2c and 2e, did the authors also check the negative expression of HOX genes expressed in other somites?

We assessed the expression of HOXC5, HOXC6/8 characteristic of cervical and brachial identity, respectively, and HOXC10 for lumbar identity. Based on the expression of the HOXC genes at the protein level, we found a predominantly cervical/brachial identity and that more posterior identities corresponding to the lumbar level were not detected. We have included the data for the reviewer below. This is expected due to the presence of retinoic acid that favors the generation of more anterior spinal cord identities (Mahony, Mazzoni et al. 2011). We are currently exploring the conditions to establish a lumbar-specific NMJ model, but this falls outside the scope of this manuscript.

Figure R3: Quantification of the number of ISLET1⁺ cells at day 20 (H1 cell line) that are positive for HOXC5, HOXC8, and HOXC10.

For the SMA model, the authors showed an obvious skeletal muscle dysplasia (Figures 5b-5d). This result is consistent with numerous reports of cellular autonomous skeletal muscle abnormalities in SMA. However, given this substantial skeletal muscle dysfunction, it is likely that the reduced muscle contractility in Figure 5f is primarily due to skeletal muscle dysfunction, which makes it difficult to assess the function of the SMA-NMJ.

While there is a significant muscle phenotype in agreement with previously published literature, the novel finding of our study is that the muscle phenotype appears *prior* to the MN phenotype. The SMA muscle maintains some contractile activity but this is entirely spontaneous. This was demonstrated by the pharmacological experiments in the previous version of the manuscript and now we amplify this evidence by optogenetic experiments included in Supplementary Fig. 8f and Video8. These data show that optogenetic stimulation was not sufficient to induce the synchronous contraction of skeletal muscle. Thus, there is also a serious connectivity abnormality, in addition to the muscle morphological abnormality.

In Supplementary Figure 5e, it is difficult to confirm that satellite cells localize underneath laminin. To show that, please provide 3D reconstructed image from a scan of the Z-axis.

We have included a 3D reconstruction Video 6 of PAX7 / Laminin to better show the localization of satellite cells.

Reviewer #3 (Remarks to the Author):

The manuscript of Urzi et al. describes the generation of a new neuromuscular junction model based on the differentiation of human pluripotent stem cells into neuromesodermal progenitors and then the capacity of these progenitors to self-organize into neuromuscular systems. The possibility to maintain these cultures up to 100 days leads to the generation of models that appear well matured and functional. The authors also demonstrate the application of this system to a neuromuscular disease, the Spinal Muscular Atrophy, that is characterized by the degeneration of spinal motoneurons.

Despite a growing number of studies describing the generation of a NMJ model from human pluripotent stem cells, the possibility to generate human NMJ model in 2D maintained for 100 days is really interesting and novel. Although the data are presented effectively, they are major concerns that required revision and mentioned below.

We would like to thank the reviewer for finding our model interesting and novel. We have performed additional experiments and added specific figures for the reviewer to address the main concerns raised. We hope that the reviewer will be satisfied with the revised version of the manuscript and figures.

1. The description of the protocol in Figure 1A as well as in the text (line 85) is not totally clear and should be improved. As indicated by the authors (line 85), « to enhance the generation of both cell types, we continued the treatment with WNT/ bFGF/2SMADi which does not correspond to the Figure 1A.

We are sorry for the confusion. We have revised Fig. 1a to better illustrate the two different differentiation protocols. The modified illustration now clearly distinguishes between the NP and NP+PSM differentiation conditions, addressing the reviewer's concern.

2. In Supplementary Figure 2b-c, could the authors confirm that all the cells TBXT positive are also SOX2 ? what is consequently the proportion of SOX2 positive cells that do not express TBXT ?

Yes, we can confirm that in the NMP conditions around 70% of the cells are TBXT⁺/SOX2⁺ and that around 30% of cells are TBX6⁺. In NMP conditions, very few, if any, cells express SOX2 in the absence of TBXT. When we add the 2SMADi with WNT/FGF we observe higher levels of SOX2 and many cells express SOX2 in the absence of TBXT (Supplementary Fig. 2b, 2c).

3. In Figure 1c, the authors suggest segregation between NP and PSM. First, what is the reproducibility of these results among the different cell lines? Could the authors also provide quantification of the proportion of PAX3 and SOX2 positive cells?

Regarding Fig. 1c, we have examined the reproducibility of the segregation between NP and PSM in the three different cell lines: H1, H9, and XM001. Our analysis demonstrates that the segregation of neural and mesodermal progenitors at day 6 is reproducible across the different cell lines. There are some differences in the propensity of each cell line to differentiate towards the neural or mesodermal lineage which we have previously observed (Martins et al., *Cell Stem Cell*, 2020). This variability between different human PSC lines is expected and also well-established in the field (Osafune et al., 2008, Bock et al., 2011, Ortmann et al., 2017). We thank the reviewer for raising this important point that we have now included in the discussion. Still, as the reviewer suggested, in the revised version of the manuscript, we have included the quantifications for PAX3 and SOX2 positive cells in all three cell lines. These quantifications further support the reproducible segregation of neural and mesodermal progenitors at day 6, which we used as an early quality control checkpoint. The data are included in Figure 1e.

4. Regarding this segregation observed at Day 6, which is a central point of this manuscript, could the authors explain why they decided to passage the cells at the same time point (day 6) as indicated line 106 ? It is indeed difficult to understand how this segregation can be maintained after dissociation. Could the authors comment?

We thank the reviewer for the comment. Indeed, the segregation at day 6 is key to this protocol and we consider that day as an early quality control checkpoint of our differentiation. At day 6 the mesodermal progenitor cells organize in dome-like structures that are prone to detachment if left unperturbed (Figure 1c). Thus, if we do not perform the passage at day 6, these dome structures start detaching from the dish, leading to the loss of the mesodermal cells. Passaging the cells at day 6 allows us to rescue the dome-like structures by replating the cells. We hypothesize that the ability of the cells to organize again after passaging is facilitated by the similar expression patterns of adhesion molecules between the neural progenitors and mesodermal progenitors at this stage. By carefully performing the passage at day 6, we can ensure the continued presence and interaction of both NP and PSM populations, which are essential for the self-organization and subsequent differentiation of the neuromuscular system in our model.

We have incorporated this explanation into the discussion to provide a clear rationale for the decision to passage the cells at day 6. Thank you for bringing up this important point, and we appreciate the opportunity to address it.

5. Could the authors also explain the choice of the concentrations used for RA and SAG? Did they optimize these concentrations as it has been previously demonstrated that the concentration of RA might influence the proportion of hPSC-derived spinal MNs (Maury et al., 2015).

We performed some preliminary experiments with different concentrations of RA and SAG and we came to the conclusion that the concentration used by Maury et al. is the best for our culture conditions. Thus, we have used the concentration suggested by Maury et al. that most efficiently specifies the motor neuron identity without compromising the differentiation potential of mesodermal cells.

6. In Figure 1d, the protocol as well as the characterization of hiPSC-derived spinal MNs are not clearly explained. What is the protocol used to generate spinal MNs? Immunostaining for TUBB3 is not adapted to evaluate the number of spinal MNs and ISL1/ILS2 or Hb9 should be considered. The same concern can be raised for the characterization of MNs obtained in the cocultured system.

We have updated the schematic illustration in Fig. 1a to make the protocol clear.

7. Line 114, could the author explain the rationale to maintain the cultures only in the presence of HGF and IGF ?

The rationale behind maintaining the cultures only in the presence of hepatocyte growth factor (HGF) and insulin growth factor (IGF) stems from the specific requirements for the survival and maturation of skeletal muscles.

During the establishment of the protocol, various growth factors were tested to determine their effects on spinal cord neurons and skeletal muscles. For example, the growth factor brain-derived neurotrophic factor (BDNF) and glial cell-derived neurotrophic factor (GDNF), known to support motor neurons, were not necessary for the survival of motor neurons in the 2D soNMJ model. This observation led to the hypothesis that motor neurons in the culture receive support from other cell types present, such as skeletal muscles and glial cells.

However, we observed that the presence of HGF and IGF played a crucial role in the proliferation of myogenic progenitors and differentiation to myofibers. Earlier studies summarized in the review by Charge & Rudnicki (Charge and Rudnicki 2004) and the work from the group of Olivier Pourquie (Chal, Oginuma et al. 2015, Chal and Pourquie 2017) have shown the importance of HGF/IGF in the differentiation of human pluripotent stem cells to skeletal muscles.

8. Regarding the characterization of the NMJ models at day 20, the authors quantified the expression of Olig2 only by RT-qPCR. Could they indicate the proportion of OLIG2 positive cells in order to determine the efficiency of DAPT treatment ? This comment is also important as from Supplementary Figure 3b, the expression of OLIG2 appears really variable among the different cell lines. Could the authors comment?

The efficiency of DAPT treatment was initially determined in the neural conditions and that concentration was then used for both conditions. We have included data for the reviewer below to show the effect of DAPT treatment by quantifying the number of OLIG2, HB9, and ISLET1 cells at day 10 (before) and day 14 (after) DAPT treatment.

Figure R4: Quantification of the number cells expressing MN progenitor markers, OLIG2 and differentiated MN markers HB9, ISL1 at day 10 and 14. Different colors refer to different cell lines: H1 (green), H9 (purple), XM001 (blue).

9. In the same line, could the author comment why ISL1 mRNA is observed only in H1 line (Supplementary Figure 3b) whereas ISL1 positive cells are also observed in XM001 line at the same time ? Did the authors look at other markers of MNs such as Hb9 ? *ISL1* is upregulated in all different cell lines at day 20 but there is some difference in the actual levels. We have included the data for all lines in Supplementary Fig. 3b. In our preliminary experiments, we couldn't observe a significant difference between the expression of HB9 and *ISL1*. For that reason, we consistently used *ISL1* for the characterization of motor neurons at day 20 and ChAT at day 50 and day 100 throughout the manuscript.

10. To determine the identity of spinal MNs, the use of HOXC6 only is not sufficient. To clearly determine the brachial identity, the authors should analyze the presence of other HOX genes, such as HOXA5 and HOXC8. Other HOX genes such as HOXD9 por HOXC10 should be analyzed in order to exclude the presence of other subtypes. The proportion of HOXC6 positive cells among the ISL1 population seems to be also very variable from 40 to 100%. This variability is also observed for the ISL1 positive cells (18,9% \pm 10.1 as indicated line 125). It is therefore difficult to understand the statement of an « efficient » generation. Could the authors comment and explain why they have such variability even with the same cell line ? It could be interesting to also indicate the proportion of neurons (by using a pan neuronal marker such TUBB3) in order to estimate the proportion of spinal MNs within the neuronal population?

We assessed the expression of HOXC5, HOXC6, HOXC8 characteristic of cervical/brachial identity and HOXC10 for lumbar identity (Fig. R3). Based on the expression of the HOXC genes at the protein level, we found a predominantly cervical/brachial identity and that more posterior identities corresponding to the lumbar level were not detected. We have included the data for the reviewer below. This is expected due to the presence of retinoic acid that favors the generation of more anterior spinal cord identities corresponding to the brachial level (Mahony et al., *Genom Biol*, 2011). We are currently exploring the conditions to establish a lumbar-specific NMJ model, but this falls outside the scope of this manuscript.

As the reviewer observed, we agree that there is a difference in the response based on the genetic background of each cell line which is well-established in the field (Osafune et al., 2008, Bock et al., 2011, Ortmann et al., 2017). However, all lines respond following the same trend from day 3 to day 6 to day 20 (Supplementary Fig.3b) and generate functional NMJs (Fig.3c, 3d).

We characterized the model as efficient based on:

The robust generation of neural and muscle progenitors (Fig. 1e) at day 6 from NMPs.

The robust generation of spinal cord neurons, skeletal muscle cells, and glia cells at day 50 and day 100 (Fig.3, Supplementary Fig.4).

The efficient generation of neuromuscular junctions at day 50 and day 100 (Fig.3d).

11. As the above comment, a better quantification of the proportion of MNs, glial cells and interneurons generated should be provided at later stage of the protocol (Day 50 and D100) ? Only examples of immunostaining are provided for the moment except for GFAP for which a quantification of the staining area is provided. Thus, a more precise quantification of the positive cells should be provided in Supplementary Figure 4. This should help at evaluating the reproducibility of the protocol. In addition, regarding the quantification of staining area, it is difficult to accurately evaluate the results as the normalization of this quantification is not indicated. The authors should indicate on which parameters they normalized this quantification ? per field ? per number of DAPI positive cells ?

We thank the reviewer for the suggestion. We have included quantifications for motor neurons and interneurons in Supplementary Fig. 5 as suggested. However, we would like to stress that such complex cultures where different cell types are generated in parallel show larger variability in the actual number of cells compared to monocultures, where we use signals that favour the generation of only one specific cell type. We have included this in the discussion.

12. Due to the variability described in Figure 2, it appears necessary to indicate if such variability is also observed at day 50 and day 100. Especially, can the beautiful and impressive image in Figure 3b be observed with the other cell lines? We thank the reviewer for finding the images impressive. We have observed the same segregation with different human PSC lines including the SMA patient cell line (Supplementary Fig. 8). This is an intrinsic property of the culture from a very early stage of differentiation due to the concomitant development of spinal cord neurons and skeletal muscle cells. We have also included an image from the XM001 iPSC cell line below for the reviewer.

Figure R5: High content imaging of a 24 well soNMJ XM001 culture stained with Fast MyHC (magenta) and TUBB3 (cyan). Scale bar 1 mm.

13. In Supplementary Figure 5d, how did the authors normalize the quantification of the NMJ length? What is the proportion of the index fusion (meaning the number of myotubes with more than 3 nuclei)? Did the authors normalize the AChR clustering with the size of myotubes?

We quantified the length of the NMJs and normalized them to the number of muscle fibers in the specific field of view. We can verify that most muscle fibers in the control lines after day 50 are multinucleated. Unfortunately, we could not establish until now an automated way to measure the number and exact size of each myotube because it is technically very challenging to segment them. For the analysis, we have counted the number of myofibers manually.

14. In figure 3D a similar concern can be raised. What does BTX/ Fast MyHC represent? the number of BTX aggregates per Fast MyHC ? did the authors select the size of BTX aggregates ? Did the authors select the Fast MyHC containing more than 3 nuclei ?

We have quantified the number of NMJs that are longer than 2 μm . We usually use longer than 5 μm but if this was applied in this study, we would lose almost all the NMJs in the SMA model where they are smaller than 5 μm (Fig. 5c). Thus, we quantified all the NMJs that are longer than 2 μm . We can confirm that at day 50 most myofibers are multinucleated in the control lines. However, it is technically too challenging to quantify the exact number of nuclei in each myofiber because they are quite long and difficult to segment using an automated way. We have quantified the number of BTX aggregates and normalized them to the total number of myofibers in the specific field. We have now included this information in the materials and method section.

15. Regarding the functionality, the authors demonstrated that the contractions can be blocked by curare treatment that act at the post-synaptic level. In order to clearly demonstrate the connectivity, the authors should demonstrate the blockage of the contractions using Botulinium Toxins as these drugs will act directly on the synaptic transmission from the MNs. We appreciate the reviewer's suggestion to use Botulinum Toxin to further validate the functionality of the neuromuscular junctions (NMJs) by directly targeting synaptic transmission from the motor neurons (MNs). However, we regret to inform you that we were unable to perform this experiment within the context of the revision due to the unavailability of the toxin on the market at the time of the revision.

However, the optogenetic experiments support the functionality of the NMJs by demonstrating the ability of the motor neurons to initiate muscle contractions upon light activation. This synchronization of muscle contractions in response to light stimulation provides strong evidence for functional connectivity between the motor neurons and skeletal muscle.

While Botulinum Toxin experiments would have provided further validation, the optogenetic experiments, along with the curare treatment results demonstrating blockage of contractions at the post-synaptic level, collectively suggest the functional integrity of the NMJs in our model. In the future, we would like to perform the botulinum toxin experiments and we thank the reviewer for the suggestion.

16. The authors claim « in those cultures, the contraction of the skeletal muscles could not be maintained for too long due to loss of attachment. An important advantage of the soNMJ model is that muscles remained attached to the dish for long periods, even under contractile conditions (Supplementary Fig. 5a,c). This is an interesting observation but could the authors comment or discuss this result? Did they observe more focal adhesions that increase muscle surface interactions ?

We speculate that the support from the neurons in the soNMJ model contributes to the longer attachment of the skeletal muscles even under contractile conditions. However, we have not performed a detailed analysis of focal adhesions in comparison to skeletal muscle cells differentiated in the absence of MNs. We agree that this is an interesting question that we would like to explore in the future.

17. At day 50 and 100, are the NMJs as mature as in neuromuscular organoids? In Faustino M et al, the group presented TEM images of synaptic clefts, do the authors have such organization in their soNMJ models?

We have included TEM images of the 2D soNMJ model at day 75. We observed that the 2D NMJs reach a level of maturation similar to the NMO level but at a slower rate. As a result, day 50 NMJs in NMOs are more similar to day 75 2D NMJs.

18. The optogenetic experiments are very elegant. However, what is the proportion of motoneurons transduced by the different AAV used for these experiments (Figure 4)?

We used the AAV9 virus serotype that infects very efficiently the human cells. We observed by fluorescence microscopy that a very high number of neurons was infected but we have not done systematic studies to quantify the numbers in each experiment because we were not fixing these cultures.

19. It could be also elegant to use optogenetic stimulation in the context of SMA. Indeed, the authors claim an early NMJ defects in SMA condition in comparison with control. The use of optogenetic will reinforce these conclusions which for the moment rely only on the quantification of NMJ.

We thank the reviewer for the suggestion. We have performed the additional optogenetic experiments in the SMA line and we have included the data in Supplementary Fig. 8f and Video 8. Indeed, optogenetic stimulation of SMA motor neurons failed to stimulate the contraction of skeletal muscles reinforcing our previous conclusions.

20. Line 284, the authors concluded that « Unexpectedly, the abundance of MNs appeared higher in SMA than in control conditions ». These results are very surprising and should be better characterized. Thus, as already mentioned for the control condition, a better quantification of the number of MNs should be determined.

We agree with the reviewer and have quantified the fold difference in the number of spinal neurons in the SMA compared to the control at day 50 in Supplementary Fig. 8c. While there is a significant reduction in the muscle area, the number of spinal neurons is not affected. It was the reduced muscle area that created the impression of increased MN abundance.

21. Similarly, a decrease in NMJ is observed but as also described by the authors SMA myotubes are smaller (line 278). This observation might explain the decrease in NMJ. Again, it should be informative to know how many skeletal muscle cells are present both in SMA and control conditions. In addition, an isogenic control should be included as strongly recommended by the ISSCR guidelines. Thus, the re-introduction of SMN gene should be performed in SMA line in order to analyze

We appreciate the reviewer's suggestion regarding the quantification of skeletal muscle cells and the inclusion of an isogenic control. We have now quantified the muscle area in both the SMA and control conditions, and the data are presented in the new Supplementary Fig. 8c on the revised manuscript.

Regarding the generation of an isogenic control by re-introducing the SMN gene into the SMA line, we understand the importance of this approach. However, targeting the SMN locus has proven to be extremely challenging due to the high repetitive genomic sequence. Despite our efforts, we have not been successful in generating an isogenic line for SMA. Generating isogenic lines for SMA has been a difficult task in the field, and currently, an isogenic line is not available for this study. However, we believe that the comparison between the SMA and control conditions still provides valuable insights into the effects of SMA on our neuromuscular junction model. We hope that future advancements in technology and methodologies will enable the generation of isogenic lines for SMA, facilitating more comprehensive analyses of the disease. We regret that this could not be achieved in the course of this project.

REVIEWER REFERENCES

- Bock, C., E. Kiskinis, G. Verstappen, H. Gu, G. Boulting, Z. D. Smith, M. Ziller, G. F. Croft, M. W. Amoroso, D. H. Oakley, A. Gnirke, K. Eggan and A. Meissner (2011). "Reference Maps of human ES and iPS cell variation enable high-throughput characterization of pluripotent cell lines." Cell **144**(3): 439-452.
- Chal, J., M. Oginuma, Z. Al Tanoury, B. Gobert, O. Sumara, A. Hick, F. Bousson, Y. Zidouni, C. Mursch, P. Moncuquet, O. Tassy, S. Vincent, A. Miyanari, A. Bera, J. M. Garnier, G. Guevara, M. Hestin, L. Kennedy, S. Hayashi, B. Drayton, T. Cherrier, B. Gayraud-Morel, E. Gussoni, F. Relaix, S. Tajbakhsh and O. Pourquie (2015). "Differentiation of pluripotent stem cells to muscle fiber to model Duchenne muscular dystrophy." Nat Biotechnol **33**(9): 962-969.
- Chal, J. and O. Pourquie (2017). "Making muscle: skeletal myogenesis in vivo and in vitro." Development **144**(12): 2104-2122.
- Charge, S. B. and M. A. Rudnicki (2004). "Cellular and molecular regulation of muscle regeneration." Physiol Rev **84**(1): 209-238.
- Lin, C. Y., M. Yoshida, L. T. Li, A. Ikenaka, S. Oshima, K. Nakagawa, H. Sakurai, E. Matsui, T. Nakahata and M. K. Saito (2019). "iPSC-derived functional human neuromuscular junctions model the pathophysiology of neuromuscular diseases." JCI Insight **4**(18).
- Mahony, S., E. O. Mazzone, S. McCuine, R. A. Young, H. Wichterle and D. K. Gifford (2011). "Ligand-dependent dynamics of retinoic acid receptor binding during early neurogenesis." Genome Biol **12**(1): R2.
- Ortmann, D. and L. Vallier (2017). "Variability of human pluripotent stem cell lines." Curr Opin Genet Dev **46**: 179-185.
- Osafune, K., L. Caron, M. Borowiak, R. J. Martinez, C. S. Fitz-Gerald, Y. Sato, C. A. Cowan, K. R. Chien and D. A. Melton (2008). "Marked differences in differentiation propensity among human embryonic stem cell lines." Nat Biotechnol **26**(3): 313-315.

REVIEWERS' COMMENTS

Reviewer #1 (Remarks to the Author):

This is a revised version of a manuscript submitted in February. The authors have extensively revised the work according to the suggestions of the reviewers. Overall, the revisions have strongly improved the work. In particular, the electron microscopic pictures have strengthened the work.

There are still a few things that need to be revised.

- Line 195: it was added that the myotubes remain attached for longer than 2D cultures due to “support that they received from the spinal cord neurons and glia cells” – this is not demonstrated; it is a hypothesis. It could also be hypothesized that the myotubes formed have a more mature signature and maybe secrete ECM proteins that enhance their attachment to the dish. This sentence should hence be rephrased (ie weakened). In the discussion, line 355, this is done correctly: “this is probably due to the continuous support that the muscle cells receive from the neurons and glia present in the soNMJ model”.
- Can the proportion of AchR clusters contacted by TUBB3+ neurites be quantified? This would help understand what proportion of the AchR clusters are actual NMJs versus clusters that have formed outside of MNs.
- Figure 3a: While the scheme is improved, it still does not reflect a mature in vivo. In the mature stage, each muscle fiber is innervated by one motor neuron; in the schematic a fiber has multiple NMJs. If these are representative of what is seen in soNMJs then it can be left this way (e.g., if you want to make the point that there is multiple innervation in the soNMJs). If not, please correct.
- Figure Legend to Figure 1: Figure 1e has no legend; in fact, the data for Fig. 1e is given as text at the end of the legend to Fig. 1d. As a consequence, the legend for Fig. f is now presented as legend for Fig. 1e. Please correct.

Reviewer #2 (Remarks to the Author):

The reviewer commends the authors for the vast amount of additional experimentation and the seriousness with which they responded to the reviewers' comments. In particular, the use of scanning electron microscopy to delineate the microstructure of the NMJ is impressive.

Unfortunately, however, the author believes that the concerns expressed in the initial review have not been addressed. The reviewer's concerns are not addressed in the present study, namely, the lack of novelty regarding 2D NMJ formation and NMJ formation by neurons and skeletal muscle cells from PSC-derived NMPs. Particularly with regard to the latter, please keep in mind that there are several previous reports of functional NMJs generated from NMPs by simultaneously culturing neurons and muscles (Martins JMF, Cell Stem Cell 2020; Pereira, JD, Nat Commun 2021). As the authors state, the rigorous

determination of culture conditions in the induction of site-specific NMPs is a crucial step, and their efforts are commendable. However, the results and the developmental potential of the method would be better discussed in a more specialized journal.

In Supplementary figure 8, instead of looking at the variation of each ROI in one well on a 96-well plate, the variation between wells should be evaluated.

Reviewer #3 (Remarks to the Author):

In this new version of the manuscript, the authors have responded effectively and satisfactorily to all the points raised by the referees, particularly those I had personally brought up. The manuscript has been significantly improved overall. Specifically, the protocol is presented in a clearer and more concise manner in this new version. Additionally, new experiments have been added to better demonstrate the reproducibility of the protocol. The results obtained in SMA are both interesting and novel, as they highlight on NMJ impairments in a human system before the death of motor neurons, a phenomenon previously demonstrated only in mouse models. However, some points still need to be clarified.

1. The results on the SMA model are very intriguing and raise many questions. These results are highly, if not overly, descriptive and would require more in-depth analyses. In particular, the authors do not appear to observe any MN death regardless of the time point analyzed. Could they comment this? Have they attempted to differentiate their SMA hiPSC lines into simple cultures of spinal Motoneurons to see if there is any mortality in this context as it has been previously described (Januel et al., 2021; Rizzo et al., 2019).

2. While the authors justify their inability to generate an isogenic control SMA hiPSC line, it is not clear why they did not use an AAV-based approach to re-express SMN? They could also investigate whether currently FDA approved treatments for SMA, such as the small molecule Risdiplam, can normalize the defects they observe”

3. In order to better define the novelty of their study, the authors should better emphasize the comparison with a recent published protocol that also described a co-differentiation protocol to convert hiPSC into skeletal innervated muscle cells with spinal motoneurons (Mazaley et al., 2020).

.Minor comments

Line 42: How are the analyses of the SMA soNMJ model high-throughput? The term “high-throughput” is misleading and suggests that high-throughput analyses have been conducted. To make such conclusion, the authors should demonstrate the feasibility of adapting their soNMJ model in 384 well plates rather than 96 well plates that is qualified of semi- highthroughput format.

Point-by-point response to reviewers' comments # NCOMMS-23-02940B

We would like to thank the reviewers for their valuable input that has improved our manuscript.

REVIEWERS' COMMENTS

Reviewer #1 (Remarks to the Author):

This is a revised version of a manuscript submitted in February. The authors have extensively revised the work according to the suggestions of the reviewers. Overall, the revisions have strongly improved the work. In particular, the electron microscopic pictures have strengthened the work.

There are still a few things that need to be revised.

- Line 195: it was added that the myotubes remain attached for longer than 2D cultures due to “support that they received from the spinal cord neurons and glia cells” – this is not demonstrated; it is a hypothesis. It could also be hypothesized that the myotubes formed have a more mature signature and maybe secrete ECM proteins that enhance their attachment to the dish. This sentence should hence be rephrased (ie weakened). In the discussion, line 355, this is done correctly: “this is probably due to the continuous support that the muscle cells receive from the neurons and glia present in the soNMJ model”.

Response: Yes, we agree with the reviewer that this is a hypothesis, and we have adjusted the text accordingly “ probably due to the support that they received from the spinal cord neurons and glia cells.”

- Can the proportion of AchR clusters contacted by TUBB3+ neurites be quantified? This would help understand what proportion of the AchR clusters are actual NMJs versus clusters that have formed outside of MNs.

Response: We would like to clarify that indeed, we have quantified only the AchR clusters that are in close contact with TUBB3⁺ neurites. We have included the explanation in the methods and clarified this in the results section: To address the reproducibility of the soNMJ model, we analyzed the number of NMJs contacted by TUBB3⁺ neurites in three different PSC lines at day 50 and day 100 (Fig. 3c, d).

- Figure 3a: While the scheme is improved, it still does not reflect a mature in vivo. In the mature stage, each muscle fiber is innervated by one motor neuron; in the schematic a fiber has multiple NMJs. If these are representative of what is seen in soNMJs then it can be left this way (e.g., if you want to make the point that there is multiple innervation in the soNMJs). If not, please correct.

Response: We have corrected the schematic and each muscle fiber is innervated by one motor neuron.

- Figure Legend to Figure 1: Figure 1e has no legend; in fact, the data for Fig. 1e is given as text at the end of the legend to Fig. 1d. As a consequence, the legend for Fig. f is now presented as legend for Fig. 1e. Please correct.

Response: We have corrected the figure legend.

Reviewer #2 (Remarks to the Author):

The reviewer commends the authors for the vast amount of additional experimentation and the seriousness with which they responded to the reviewers' comments. In particular, the use of scanning electron microscopy to delineate the microstructure of the NMJ is impressive.

Unfortunately, however, the author believes that the concerns expressed in the initial review have not been addressed. The reviewer's concerns are not addressed in the present study, namely, the lack of novelty regarding 2D NMJ formation and NMJ formation by neurons and skeletal muscle cells from PSC-derived NMPs. Particularly with regard to the latter, please keep in mind that there are several previous reports of functional NMJs generated from NMPs by simultaneously culturing neurons and muscles (Martins JMF, Cell Stem Cell 2020; Pereira, JD, Nat Commun 2021). As the authors state, the rigorous determination of culture conditions in the induction of site-specific NMPs is a crucial step, and their efforts are commendable. However, the results and the developmental potential of the method would be better discussed in a more specialized journal.

Response: We thank the reviewer for recognizing the number of additional experiments that we performed to address the specific concerns raised. Indeed, my lab previously published the generation of functional NMJs from NMPs in 3D. Other labs have also generated 3D models that include spinal cord neurons and muscle cells by self-organization or assembly. Still, this is the first study to demonstrate the generation of a self-organizing functional NMJ model in 2D through a neuromesodermal progenitor intermediate state that acquires a specific positional identity.

In Supplementary Figure 8, instead of looking at the variation of each ROI in one well on a 96-well plate, the variation between wells should be evaluated.

Response: We are sorry for the misunderstanding. This is exactly what we have plotted, the individual wells, as the reviewer suggested. We have clarified this point in the figure legend.

Reviewer #3 (Remarks to the Author):

In this new version of the manuscript, the authors have responded effectively and satisfactorily to all the points raised by the referees, particularly those I had personally brought up. The manuscript has been significantly improved overall. Specifically, the protocol is presented in a clearer and more concise manner in this new version. Additionally, new experiments have been added to better demonstrate the reproducibility of the protocol. The results obtained in SMA are both interesting and novel, as they highlight on NMJ impairments in a human system before the death of motor neurons, a phenomenon previously demonstrated only in mouse models. However, some points still need to be clarified.

1. The results on the SMA model are very intriguing and raise many questions. These results are highly, if not overly, descriptive and would require more in-depth analyses. In particular, the authors do not appear to observe any MN death regardless of the time point analyzed. Could they comment this? Have they attempted to differentiate their SMA hiPSC lines into simple cultures of spinal Motoneurons to see if there is any mortality in this context as it has been previously described (Januel et al., 2021; Rizzo et al., 2019).

Response: The absence of an early motor neuron phenotype in our complex model, where motor neurons co-develop with other cell types, underscores the significance of utilizing models that closely mirror developmental processes. While we acknowledge the reviewer's suggestion to explore simple monocultures or cocultures as potential avenues, we think that these approaches might not capture the dynamic interactions seen in our complex self-organizing model. Thus, the decision not to incorporate such comparisons in the current study stems from the scope of our investigation, which centers on the establishment of the self-organizing neuromuscular junction (NMJ) model.

2. While the authors justify their inability to generate an isogenic control SMA hiPSC line, it is not clear why they did not use an AAV-based approach to re-express SMN? They could also investigate whether currently FDA approved treatments for SMA, such as the small molecule Risdiplam, can normalize the defects they observe”

Response: We agree with the reviewer that these are open questions that we could address in the future with the established model. However, we think that these studies will take a considerable amount of time and fall outside the scope of the specific manuscript.

3. In order to better define the novelty of their study, the authors should better emphasize the comparison with a recent published protocol that also described a co-differentiation protocol to convert hiPSC into skeletal innervated muscle cells with spinal motoneurons (Mazaleyrat et al., 2020).

Response: Indeed, Mazaleyrat et al, have described a protocol to generate skeletal muscle cells from human iPSCs. During the differentiation of iPSCs to muscles, the authors observed the appearance of motor neurons. However, in their protocol, the developmental origin of the neural cells is not discussed. Based on the data presented in that manuscript, there is no evidence of the generation of an early neuromesodermal progenitor state in the specific protocol.

Minor comments

Line 42: How are the analyses of the SMA soNMJ model high-throughput? The term “high-throughput” is misleading and suggests that high-throughput analyses have been conducted. To make such conclusion, the authors should demonstrate the feasibility of adapting their soNMJ model in 384 well plates rather than 96 well plates that is qualified of semi-highthroughput format.

Response: There are many examples in the literature where 96 well plates have been used for high throughput studies and thus, we don't think that our term is misleading.

Still, we would like to emphasize that scaling from 96 wells to 384 well plates is not a problem and can be achieved by adjusting the number of cells in the 2D model.